# Rational design of a microbial consortium of mucosal sugar utilizers reduces *Clostridiodes difficile* colonization

Fátima C. Pereira [1], Kenneth Wasmund[1], Iva Cobankovic[2], Nico Jehmlich [3], Craig W. Herbold[1], Kang Soo Lee [4,5], Barbara Sziranyi[1], Cornelia Vesely[6], Thomas Decker [7], Roman Stocker [4,5], Benedikt Warth [2], Martin von Bergen[3], Michael Wagner[1,8] & David Berry [1,9]✉

Many intestinal pathogens, including *Clostridioides difficile*, use mucus-derived sugars as crucial nutrients in the gut. Commensals that compete with pathogens for such nutrients are therefore ecological gatekeepers in healthy guts, and are attractive candidates for therapeutic interventions. Nevertheless, there is a poor understanding of which commensals use mucin-derived sugars in situ as well as their potential to impede pathogen colonization. Here, we identify mouse gut commensals that utilize mucus-derived monosaccharides within complex communities using single-cell stable isotope probing, Raman-activated cell sorting and mini-metagenomics. Sequencing of cell-sorted fractions reveals members of the underexplored family Muribaculaceae as major mucin monosaccharide foragers, followed by members of Lachnospiraceae, Rikenellaceae, and Bacteroidaceae families. Using this information, we assembled a five-member consortium of sialic acid and N-acetylglucosamine utilizers that impedes *C. difficile*'s access to these mucosal sugars and impairs pathogen colonization in antibiotic-treated mice. Our findings underscore the value of targeted approaches to identify organisms utilizing key nutrients and to rationally design effective probiotic mixtures.

[1] University of Vienna, Centre for Microbiology and Environmental Systems Science, Department of Microbiology and Ecosystem Science, Althanstrasse 14, 1090 Vienna, Austria. [2] University of Vienna, Faculty of Chemistry, Department of Food Chemistry and Toxicology, Währinger Straße 38, 1090 Vienna, Austria. [3] Helmholtz-Centre for Environmental Research - UFZ, Department of Molecular Systems Biology, Permoserstraße 15, 04318 Leipzig, Germany. [4] Ralph M. Parsons Laboratory for Environmental Science and Engineering, Department of Civil and Environmental Engineering, Massachusetts Institute of Technology, Cambridge, MA, USA. [5] Institute for Environmental Engineering, Department of Civil, Environmental and Geomatic Engineering, ETH Zurich, Zurich, Switzerland. [6] Medical University of Vienna, Center for Anatomy and Cell Biology, Division of Cell and Developmental Biology, Vienna, Austria. [7] Max F. Perutz Laboratories, Department of Microbiology, Immunobiology and Genetics, University of Vienna, Vienna, Austria. [8] Center for Microbial Communities, Department of Chemistry and Bioscience, Aalborg University, 9220 Aalborg, Denmark. [9] Joint Microbiome Facility of the Medical University of Vienna and the University of Vienna, Vienna, Austria. ✉email: david.berry@univie.ac.at

Host-derived compounds constitute a reliable nutrient source for gut microbes[1–3], and can be especially important for sustaining members of the gut microbiota when diet-derived nutrients become scarce[4–6]. The mucus layer that covers the mammalian intestinal epithelium constitutes a source of carbon and nitrogen for gut bacteria[7–10]. Highly glycosylated mucins are the dominant structural component of mucus, with O-glycan carbohydrates making up to 80% of the total mucin mass[11]. Secreted mucin O-glycans have different core structures composed of N-acetylgalactosamine (GalNAc), N-acetylglucosamine (GlcNAc), and galactose residues, which can be further extended with galactose, GlcNAc, GalNAc, fucose or sialic acid (N-acetylneuraminic acid; NeuAc or Neu5Ac) sugar residues, with the latter two frequently occupying terminal positions[12–15] (Fig. 1a). The ability of gut bacteria to degrade whole mucin O-glycan structures has been characterized in cultivated gut organisms including representative species from *Bacteroides, Bifidobacterium, Akkermansia* and *Ruminococcus*[14]. Genomic analysis of available human gut bacterial genomes has predicted that 86% of the 397 analyzed organisms possess the capability to cleave, and 89% to catabolize, at least one of the mucin O-glycan monosaccharides[14,15]. Therefore, the genomic potential to catabolize these compounds is widespread among genomes of gut organisms. However, it is not known which of the organisms harboring the necessary catabolic pathways are key mucosal sugar foragers within the context of natural gut microbial communities, where complex ecological processes may influence the use of different substrates by distinct organisms. Further, it is unclear whether preferences towards particular monosaccharides in O-glycans exist among commensal gut members, as is the case for some pathogens.

In addition to commensals, the ability to utilize mucin glycans or particular mucin-derived monosaccharides as a source of nutrients confers a competitive advantage to several gut pathogens[16–19]. *Clostridioides difficile* is an important nosocomial pathogen that benefits from host-derived compounds, such as the mucosal sugars NeuAc and GlcNAc, to colonize the gut and cause severe intestinal infections[17,19–22]. *C. difficile* is also able to catabolize another mucosal sugar, galactose, though galactose utilization has not been implicated as an important factor for *C. difficile* colonization[19,22]. Depletion of the indigenous microbiota by antibiotic treatment is the major risk factor for *C. difficile* infection (CDI)[23,24], and microbiota restoration via fecal microbiota transplantation is currently the most successful way of treating CDI[25]. Mechanisms through which the indigenous microbiota provides resistance to *C. difficile* colonization include: (i) production of inhibitory substances such as thuricidin or reuterin[26–28], (ii) transformation of bile acids[29,30], and (iii) depletion or transformation of host and diet-derived nutrients[17,21,31,32]. Individual strains or a mixture of four strains that provide partial restoration of colonization resistance to CDI via some of the mechanisms described above have been identified[28–30,33]. However, no bacteriotherapy mixture able to limit *C. difficile* use of mucosal sugars has been documented to date.

Here we employ deuterium (D) stable isotope probing combined with Raman-activated cell sorting (RACS) and metagenomics to identify organisms that can forage on O-glycan monosaccharides in the mouse gut. This approach allows us to non-destructively track microbial activity at the single-cell level in response to a variety of compounds, thereby enabling post-measurement molecular analysis, and has been successfully employed in the past to identify foragers of host-derived proteins from the mouse gut[34,35]. Applying this recently developed technique, we identified a number of bacterial taxa as generalists in regard to mucosal sugar foraging, being able to utilize all the different O-glycan sugars, as well as taxa that seem to be specialized on only a subset of the available mucosal sugars. With this new information, we designed a bacteriotherapy mixture based on the organisms identified as active NeuAc and GlcNAc foragers and evaluated the impact of their presence on *C. difficile* growth in mice. We show that this bacterial consortium is able to decrease the availability of these mucosal sugars and can reduce *C. difficile* growth both in vitro and in vivo. Our approach therefore identified a consortium of gut bacteria that contributes to colonization resistance against *C. difficile*, and further indicates that depletion of mucosal sugars is one of the mechanisms underpinning this resistance.

## Results

**Tracking microbial activity in response to mucin O-glycan monosaccharides.** To investigate the utilization of mucin O-glycan monosaccharides (NeuAc, GlcNAc, fucose, galactose, and GalNAc) by colonic microbiota, freshly collected mouse colon microbiota samples were incubated ex vivo with different concentrations of each of the mucosal O-glycan monosaccharides individually, or mucin itself, in the presence of 50% heavy water ($D_2O$) under anaerobic conditions at 37 °C (Fig. 1a). Based on a preliminary experiment to determine the most suitable time point to efficiently probe for D incorporation, an incubation period of 6 h was selected for all subsequent incubations (Supplementary Fig. 1). Final concentrations of different mucosal monosaccharides supplemented to the microcosms were in the low millimolar range (between 3 and 28 mM; Fig. 1c bottom panel). The lowest concentration values were selected based on reported concentrations of the different monosaccharides in purified hog gastric mucin and mucin gels[36,37] and on the typical O-glycan structure of the large intestine, i.e., sugars such as galactose and GlcNAc are more abundant than are GalNAc or the terminal residues fucose and NeuAc[38]. Cells stimulated by each supplemented monosaccharide become active and as a result incorporate D into cellular constituents such as lipids or proteins, which can be effectively detected by single-cell Raman microspectroscopy[34]. This approach was applied to cells from supplemented microcosms of three independent experiments (MonoA, MonoB, and MonoC; see "Methods"). Amendment of all mucosal sugars (in separate experiments) resulted in activity-induced D incorporation (measured as %CD[34]) into cells above threshold levels, while cells from control microcosms that were exposed to deuterated water but not supplemented with sugars showed only negligible levels of D incorporation (Fig. 1b, c). The lowest percentage of cells showing D incorporation was observed for amendments with 3 mM fucose (9.4 ± 2.9%) and the highest for cells supplemented with 28 mM of galactose (73.8 ± 8.5%) (Fig. 1b, c). When D incorporation levels were compared among treatments with similar concentrations of each compound (5-6 mM), galactose resulted in D incorporation by the largest fraction of cells within the community (62.5 ± 5%; $p = 0.005$, ANOVA) in comparison to the other sugars (Fig. 1c, inset chart). These results are in agreement with the fact that a larger number of gut organisms encode enzymatic pathways for galactose catabolism, with NeuAc, fucose and GalNAc catabolism pathways being less prevalent among gut bacteria[15]. Quantification of mucosal sugars in the microcosms revealed that microbial-mediated conversion led to depletion of a significant fraction (12–70%) of the monosaccharides when compared to abiotic controls ($p = 4.809e−06$, Mann–Whitney test; Fig. 1d). Nevertheless, we observed no correlation between the fraction of sugar consumed and the levels of D incorporation (%CD) or the percentage of D-labeled cells (Fig. 1b–d). This suggests that other factors such as the amount of energy gained from the catabolism

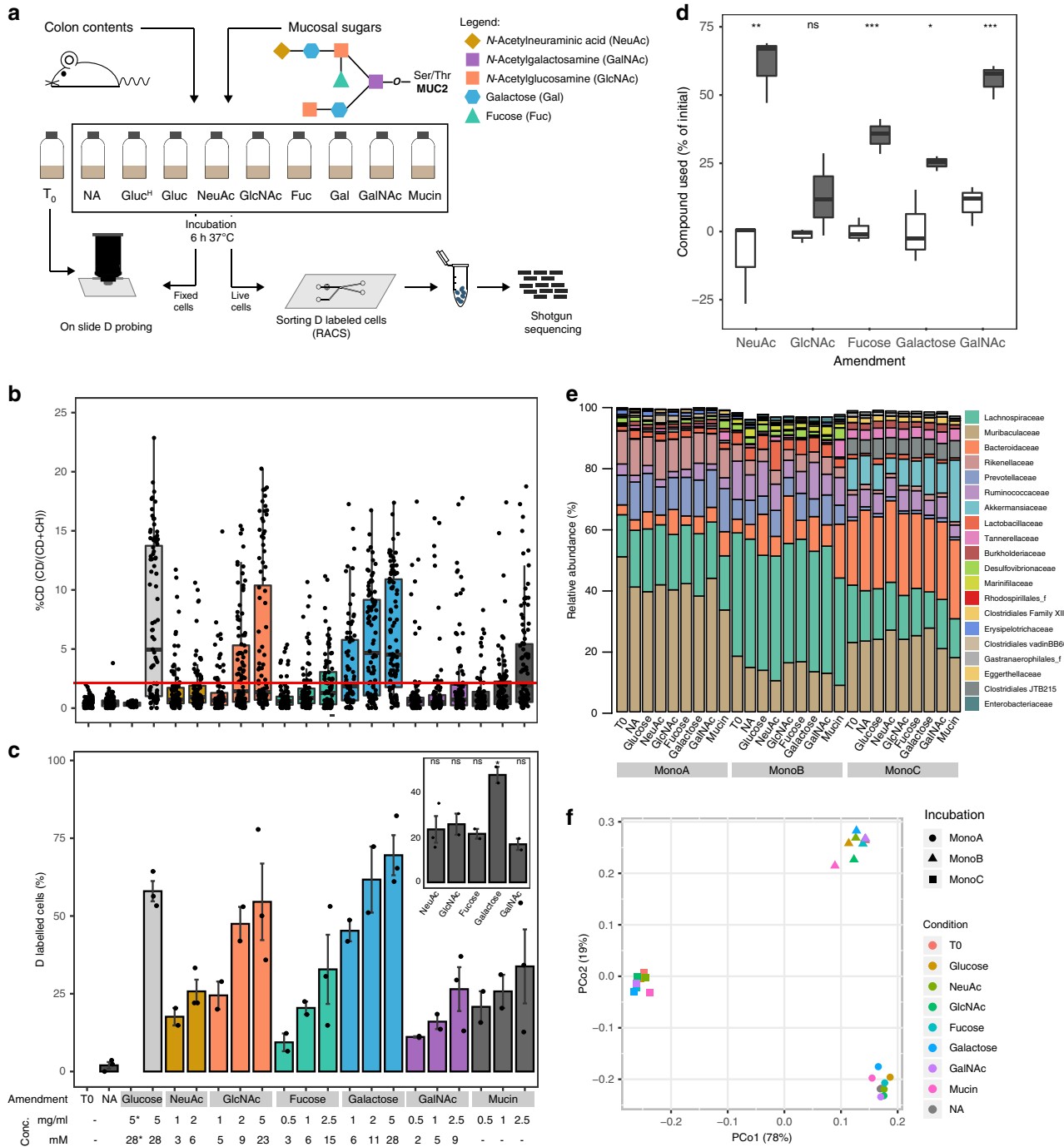

of each of these compounds, the different physiologies of the taxa involved in their degradation and/or possible conversion of sugars into storage compounds may play an important role in their biosynthetic activities.

In an attempt to gain initial insights into the identities of the active cells, we analyzed initial and supplemented microcosms by 16S rRNA gene amplicon sequencing over-time. Members of the phyla Bacteroidota and Firmicutes, and of the families Lachnospiraceae, Muribaculaceae (formerly classified as Bacteroidales S24-7 family) and Bacteroidaceae dominated the initial bacterial community in our microcosm experiments and no significant shifts in the community composition were detected during incubation (Fig. 1e, f). Rather, the community composition cluster by experiment ($p < 0.001$, $r^2 = 0.132$, PERMANOVA; Fig. 1f), which can be attributed to differences in the composition

of the mouse colon microbiota contents used in different incubations. Furthermore, no significant clustering of the bacterial community by sugar supplemented was observed, indicating that the short incubation time used was able to limit the expansion of stimulated organisms (Fig. 1f). Because activity measurements using D and Raman spectroscopy do not require cell division[34], this approach allowed us to conclude that a significant part of the mouse colon microbiota uptakes, and as result becomes active, upon amendment with different mucosal O-glycan monosaccharides, with response to galactose being the most prominent.

**Sorting, sequencing, and identification of O-glycan monosaccharide utilizers.** In order to identify the organisms involved in mucin O-glycan foraging, we sorted non-fixed D-labeled

**Fig. 1 Application of $D_2O$ to identify active mucin monosaccharide utilizers. a** Experimental setup. Mouse colon contents were diluted in PBS and supplemented with indicated compounds; $T_0$ refers to microcosms processed immediately after establishment. After an incubation period of 6 h, cells from all microcosms were processed as described in the "Methods" section, and subsequently probed for D incorporation by Raman microspectroscopy or sorted based on their incorporated D levels by RACS. Sorted cells were processed for sequencing. **b** Deuterium incorporation (measured by %CD) of randomly selected cells. Boxes represent median, first and third quartile. Whiskers extend to the highest and lowest values that are within one and a half times the interquartile range. **c** Average percentage of cells labeled (i.e., with %CD higher than threshold) per microcosm. Bars represent average and standard deviation of independent experiments. The red horizontal line at 2.36 %CD indicates the threshold for considering a cell labeled. It was determined by calculating the mean + 3SD of %CD in randomly selected cells from sample Gluc[H], in which cells were incubated without addition of heavy water. Different concentrations of compound supplemented to each microcosms are shown below the horizontal axis (**c**). Asterisk indicates the Gluc[H] sample. Insert chart shows a direct comparison of the percentage of labeled cells in microcosms supplemented with comparable concentrations of each mucosal monosaccharide (5–6 mM) (*$p = 0.005$; one-way ANOVA). All cells measured from two independent experiments are depicted in (**b**) and average percentages from both experiments are depicted in (**c**). **d** Percentage of supplemented mucosal monosaccharide utilized in three independent microcosms per condition, established with live (gray boxes) or autoclaved (white boxes) biomass. Initial and final (after 6 h) concentrations of compounds were quantified using HILIC for microcosms supplemented with the highest concentration of each compound tested in (**c**). Boxes represent median, first and third quartile. Whiskers extend to the highest and lowest values that are within one and a half times the interquartile range (*$p = 0.0360$; **$p = 0.0038$; ***$p = 0.0008$ for galactose, NeuAc and fucose, respectively, or ***$p = 0.0006$ for GalNAc; ns=non-significant; two-tailed Welch two-sample $t$ test). **e** Initial microcosms composition (family level) determined by 16S rRNA gene amplicon sequencing from three independent experiments (MonoA, B and C). **f** Variation between samples (Principal Coordinates Analysis based on Bray–Curtis dissimilarities) colored by the monosaccharide supplemented for 3 independent experiments. For each experiment, the initial time point before supplement (T0) and the final time point after incubation (6 h) with each supplemented monosaccharide are represented. The percentage in brackets correspond to the variation explained by each principal coordinate (PCo) to the fraction of the total variance of the data. Source data are provided as a Source Data file.

cells from the supplemented microcosms using RACS and shotgun-sequenced DNA from collected cells (Supplementary Fig. 2c,d). Between 125 and 244 cells were sorted for each amendment, yielding the recovery of 23–43 metagenome-assembled genomes (MAGs) per amendment (Fig. 2a, Supplementary Table 1). No MAGs were recovered from sorts from two negative controls consisting of microcosms incubated with glucose in the absence of D (Gluc[H]). MAGs represented a total of 51 unique population genomes (as defined by sharing an average nucleotide identity[39], ANI < 99%), including 9 near-complete genomes and 24 substantially complete genomes (Supplementary Data 1). They belonged predominantly to the phyla Bacteroidota and Firmicutes, with a small number belonging to the phyla Verrucomicrobiota, Proteobacteria, Desulfobacterota and Patescibacteria (Fig. 2a, Supplementary Fig. 3, Supplementary Data 1). A large percentage (43%) of the retrieved MAGs belonged to the family Muribaculaceae, a result supported by previous studies that identified members of this family as potential mucus degraders[34,35,40]. Several organisms were recovered from amendments with at least four out of the five individual monosaccharides, as well as from mucin amendments (11 top MAGs; Fig. 2a). We defined these taxa as "mucosal sugar generalists" (further discussed in the "Discussion" section).

To discard the possibility that the pool of generalist organisms we identified was not simply influenced by the high relative abundances of these organisms in the microcosms, we determined the initial relative abundance of sorted organisms prior to sorting. Metagenomic sequencing of initial microcosms allowed us to retrieve 120 MAGs (named "MMAGs" for Microcosm MAG; Supplementary Data 1), of which, 44 overlapped with RACS-retrieved MAGs (ANI > 99%) (Supplementary Fig. 3, Supplementary Table 2). Calculation of relative abundances (based on read coverage) of generalist organisms revealed that 5 out of 11 were present at low relative abundances (below 1%) in the initial microcosms (Fig. 2a). Mucosal-sugar generalists identified here belonged almost exclusively to the order Bacteroidales or to the family Lachnospiraceae, supporting previous predictions from genomic analyses that reported Bacteroidetes and Clostridia as major mucin generalists from the human gut[15].

Enzymatic hydrolysis of different mucosal sugars, as well as canonical pathways for catabolism of cleaved monosaccharides,

have been described and identified for a subset of human gut microbes[15]. To further evaluate the success of our approach in selecting organisms involved in mucosal sugar utilization upon mucin breakdown, sorted fractions as well as the unsorted initial microcosms were queried for the presence of genes encoding enzymes involved in mucosal sugar uptake[35] (Supplementary Table 3). We detected an enrichment of up to four-fold for genes involved in mucin degradation in the sorted fractions compared with unsorted metagenomes. Overall, the screened genes were twice as abundant in sorted fractions ($P = 0.0247$, one-sample $t$ test), and all the genes screened were enriched in these fractions with the sole exception of an alpha-galactosidase (Supplementary Table 3).

We further analyzed the genomic potential of the individual sorted organisms identified by RACS to catabolize different O-glycan sugars. To this end, we screened all sorted MAGs for the presence of enzymes involved in the hydrolysis and catabolism of the respective sugar monosaccharides (i.e., each MAG was screened for a specific catabolic pathway only if it was recovered from amendments of that particular monosaccharide: 108 MAG-monosaccharide combinations were screened). Despite the fact that retrieved MAGs consisted of incompletely reconstructed genomes, we could identify complete or near-complete catabolism pathways (with the exception of GalNAc) in many (48%) of the MAG-monosaccharide combinations. For 82% of the MAG-monosaccharide combinations screened, we identified at least one enzyme involved in the catabolism and/or cleavage of the respective monosaccharide (Fig. 2b; Supplementary Data 2), though we could not identify any organisms that encoded enzymes for a complete GalNAc catabolism pathway in our dataset. Further, the large majority of MAGs (19 out of 25) recovered from GalNAc amendments do not encode any of the known enzymes necessary for the canonical catabolism of this monosaccharide, notwithstanding the presence of cleavage enzymes (Fig. 2b; Supplementary Data 2).

**Cross-validation of D incorporation into RACS-sorted taxa using SIP-metaproteomics.** To corroborate the D-labeling of organisms sorted and identified by RACS, we performed metaproteomic analyses to identify incorporation of D into peptides produced by members of the gut microbiota[41]. To this end, we

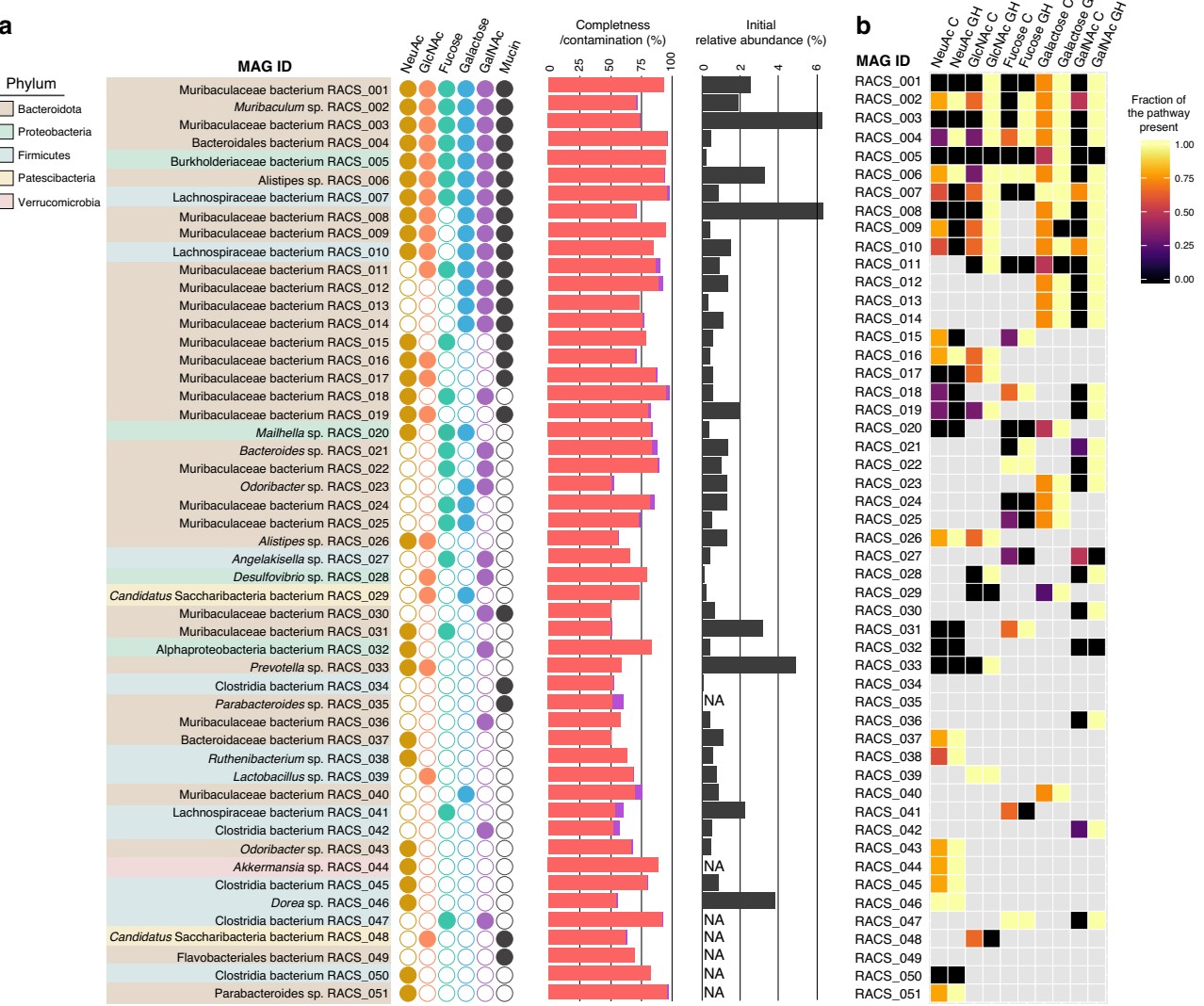

**Fig. 2 Targeted mini-metagenomics of monosaccharide-utilizing bacteria from the mouse colon microbiota. a** Identity of all MAGs recovered from sequencing of RACS-sorted fractions. Color shades denote the bacterial phylum to which the MAG belongs. Filled circles denote from which amendment each MAG has been recovered. MAGs recovered from different sorted fractions but with an ANI > 99% were considered to be the same, and a representative was selected based on dRep analysis[62]. Completeness (red bars) and contamination (purple bars) calculated with CheckM[60] are shown for each representative MAG. The relative abundance (calculated based on metagenome coverage) of each MAG in the initial microcosms is indicated by gray bars. For some RACS-MAGs relative initial abundances could not be calculated (not available-NA) because no corresponding MMAG was retrieved from the microcosms metagenome. **b** genome-encoded O-glycan-cleavage and catabolic capability of monosaccharide-stimulated RACS organisms. Heatmap showing the percentage of enzymes involved in the hydrolysis (glycoside hydrolases: GH) and the completeness of known catabolic pathways (catabolism enzymes: C) of each compound (Supplementary Data 2). Only MAG-monosaccharide combinations that were recovered by RACS-sorting are depicted. MAG-monosaccharide combinations that have not been recovered are shown in gray. Source data are provided as a Source Data file.

analyzed peptides from all microcosms using high-resolution mass spectrometry and quantified the relative isotope abundance (RIA) of D in retrieved peptides. A custom-made database consisting of protein-encoding sequences recovered from the MMAGs was used for peptide identification. Between 6,808 and 16,676 total unique peptides were identified from different supplemented microcosms (Supplementary Table 4). Of these, a low but significant percentage was shown to have incorporated D compared to no-amendment controls (Fig. 3a). The lowest percentage of identified peptides with incorporated D ($0.36 \pm 0.03\%$) was detected for NeuAc-supplemented microcosms and the highest ($3.1 \pm 2.5\%$) for mucin-supplemented microcosms (Fig. 3a). The majority of the D-labeled peptides matched proteins with housekeeping functions and/or which were involved in central metabolism pathways, though we also detected D

incorporation into peptides matching O-glycan degradation-specific proteins (Supplementary Table 5). Similar average levels of incorporated D into labeled peptides were found across all supplemented microcosms, which may be due to the short incubation times used in our experiments (Fig. 3b). Members of the Lachnospiraceae, Muribaculaceae, Bacteroidaceae and Rikenellaceae families dominated total metaproteomes (i.e., unlabeled and D-labeled peptides) of supplemented microcosms. The relative abundance levels of peptides from these groups strongly depended on the experiment (Mono-A, -B, or -C) (Fig. 3c), which is in agreement with 16S rRNA amplicon sequencing analysis (Fig. 1f). Importantly, the family level profiles of D-labeled peptides differed from that of the total identified peptides, and were dominated by members of the family Muribaculaceae (Fig. 3d). These results were therefore in good agreement with our

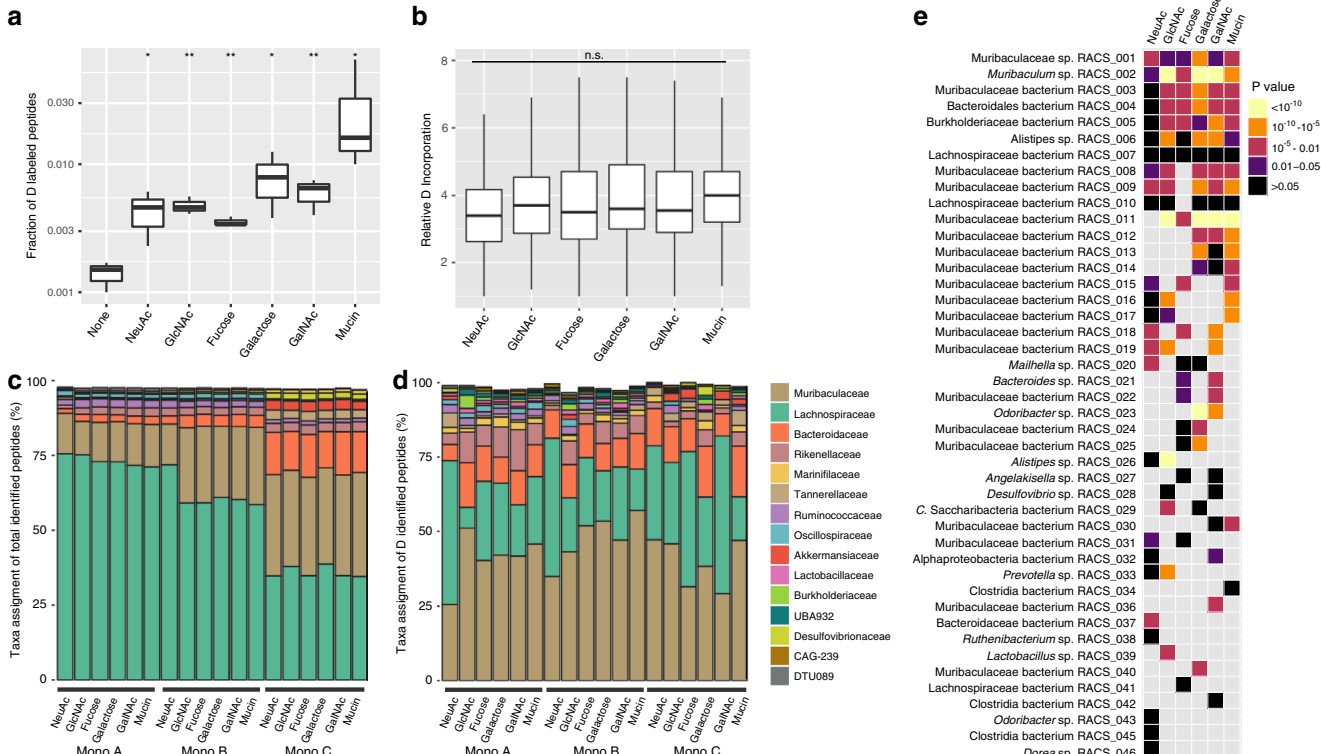

**Fig. 3 Deuterium incorporation into peptides from *O*-glycan monosaccharide-supplemented microcosms. a** Fraction of peptides with incorporated deuterium per amendment obtained by dividing the number of peptides with incorporated D by the total number of peptides. Significantly higher percentages are found for supplemented microcosms compared with the no-amendment control (*$p = 0.030$ for NeuAc; *$p = 0.028$ for galactose; *$p = 0.049$ for mucin; **$p = 0.001$ for GlcNAc and fucose; **$p = 0.006$ for GalNAc; one-tailed Welch two-sample $t$ test using "None" as a reference group). **b** Relative D abundance of D-labeled peptides from three different microcosms per condition ($p = \text{ns} = $ non-significant; One-way ANOVA). In (**a**) and (**b**) boxes represent median, first and third quartile. Whiskers extend to the highest and lowest values that are within one and a half times the interquartile range. **c, d** Taxa assignment of total unique peptides (**c**) or deuterium labeled peptides (**d**) at the family level identified by metaproteomics analysis. Taxonomic data from microcosms from three independent experiments is shown. **e** Heatmap showing the $p$-value significance level of enrichment of each MMAG (equivalent to each RACS-MAG) in the D peptide pool compared to the total peptide pool (two-proportion Z-test). Only MAG-monosaccharide combinations recovered by RACS-sorting are depicted. MAG-monosaccharide combinations that have not been recovered are shown in gray. Source data are provided as a Source Data file.

RACS-derived results, in which the most common sorted organisms belonged to the Muribaculaceae (Fig. 2a). Furthermore, by comparing the distribution of peptides from the different MAGs in the D-labeled peptide fraction with the total peptides, we observed that peptides belonging to sorted MAGs had comparatively higher contributions to the D-labeled peptide pool than the total peptide pool. In 68% of the cases, we could detect an enrichment of peptides from a particular MAG in the D-labeled peptide fraction of the corresponding treatment from which it was recovered (Fig. 3e; $p < 0.05$, Two-proportion Z-test). Of note, 72% (18 out of 25) of the MAGs recovered from GalNAc amendments were enriched in the D-labeled peptide pool compared to the total peptide pool (Fig. 3e), despite that fact that none of these MAGs encoded enzymes for canonical pathways for catabolism of GalNAc (see "Discussion"). These results confirm that the organisms retrieved by RACS were indeed D-labeled.

**Depletion of NeuAc and GlcNAc by selected commensals reduces *C. difficile* growth in vitro.** The pathogen *C. difficile* upregulates genes encoding enzymes involved in the uptake of NeuAc and GlcNAc during gut colonization[19,21,22]. In line with these findings, such facilitated access to NeuAc increases *C. difficile* colonization levels in vivo, and genetic inactivation of the

NeuAc transporter is associated with impaired pathogen colonization[17]. These observations prompted us to investigate whether commensal NeuAc and GlcNAc utilizers identified here could efficiently deplete these mucosal sugars, and by doing so, reduce *C. difficile* growth and/or colonization. We selected a bacterial consortium among identified NeuAc and/or GlcNAc utilizers based on the following premises: (i) their genomes contain complete GlcNAc and/or NeuAc catabolism pathways (Supplementary Data 2); (ii) a pure culture is available (or that of a closely related organism; Supplementary Table 2); and (iii) the selected organisms would cover as much as possible the phylogenetic diversity of the GlcNAc and NeuAc utilizers identified (i.e., cover 3 out of the 5 phyla identified by sorting; Supplementary Fig. 3). We therefore selected a 5-member consortium ("BacMix" for bacteriotherapy mixture) that includes the following organisms: *Akkermansia muciniphila* type strain Muc, *Ruthenibacterium lactatiformans* type strain 585-1, *Alistipes timonensis* type strain JC136, *Muribaculum intestinale* type strain YL27, and *Bacteroides* sp. isolate FP24 (Supplementary Fig. 3; see Methods).

To address if the BacMix consortium was able to compete with *C. difficile* for NeuAc and GlcNAc, we grew *C. difficile* in a diluted rich medium without (A II medium) or with (A II medium+ sugars) NeuAc and GlcNAc as carbon/energy sources, and evaluated the impact of pre-adding the BacMix consortium to this

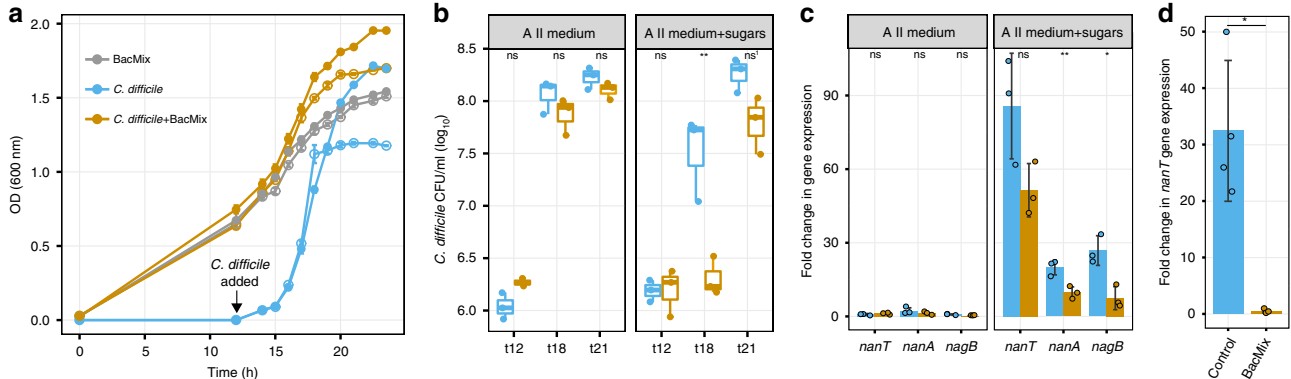

**Fig. 4 Targeted impairment of *C. difficile* growth by the BacMix consortium is accompanied by a decreased expression of NeuAc and GlcNAc catabolism genes. a** Optical density (OD) measurements of a BacMix culture, of a *C. difficile* culture or of a BacMix-*C. difficile* co-culture grown in diluted A II medium supplemented (filled symbols) or not (open symbols) with NeuAc and GlcNAc. The arrow denotes the time of *C. difficile* inoculation (12 h). **b** *C. difficile* CFU per ml of mono-culture (blue) or in a BacMix-*C. difficile* co-culture (brown) in medium non-supplemented (left panel) or supplemented (right panel) with sugars. Boxes represent median, first and third quartile and whiskers represent minimum and maximum values from three different growth vials per condition (**$p = 0.0058$, ns=non-significant; ns[1]: $p = 0.0516$; two-tailed Welch two-sample $t$ test). **c** Induction of *C. difficile* nanA, nanT and nagB expression in three independent growth vials collected at t18 from *C. difficile* mono cultures (blue) or BacMix-*C. difficile* co-cultures (brown) in diluted A II medium not supplemented (left panel) or supplemented (right panel) with sugars, relative to growth in minimal medium containing 0.5% glucose (*$p = 0.0118$, **$p = 0.0047$, ns=non-significant; two-tailed Welch two-sample $t$ test). **d** *C. difficile* nanT expression in fecal samples 2 days post-infection of antibiotic-treated control (blue) or and BacMix-recipient (brown) mice relative to growth in minimal medium containing 0.5% glucose ($n = 4$/group) (*$p = 0.0021$; two-tailed Welch two-sample $t$ test). In (**c**, **d**), bars represent average and standard deviation and each data point is represented as a single dot. Source data are provided as a Source Data file.

medium on *C. difficile* growth (Fig. 4a). This medium was able to support growth of all BacMix strains and of *C. difficile* independently, with the exception of *R. lactatiformans* (Supplementary Fig. 4). Importantly, addition of GlcNAc and/or NeuAc to the medium boosted bacterial growth so that higher final optical densities (OD) were reached for all growing strains (Supplementary Fig. 4). These results showed that the BacMix members are efficient GlcNAc and/or NeuAc utilizers and that these sugars are indeed able to sustain their growth. Strikingly, significantly lower *C. difficile* titers were determined when the BacMix consortium was pre-added to the medium containing GlcNAc and NeuAc (Fig. 4b; right panel; t18: $p = 0.0058$, t21: $p = 0.0516$, Welch two-sample $t$ test), though no significant reduction due to the BacMix addition could be detected in the absence of the sugars (Fig. 4b, left panel). This suggests that exhaustion of these particular nutrient sources, but not of other components of the basal medium, is able to impact *C. difficile* expansion. To further test this hypothesis, we quantified the expression of *C. difficile* genes previously shown to be critical for the catabolism of NeuAc and/or GlcNAc (nanT, nagA, and nagB; targeting a NeuAc transporter, an *N*-acetylneuraminate lyase and an *N*-acetylglucosamine-6-phosphate deacetylase, respectively[17,19]) during *C. difficile* growth in vitro by RT-qPCR at mid-exponential phase (t18). As expected, only very low expression levels of these genes were detected (relative to growth on diluted A II medium supplemented with glucose) when *C. difficile* was grown in the absence of sugars (Fig. 4c, left panel: A II medium). Increased gene expression was observed after sugar amendment, but this induction was significantly reduced for all three genes in the presence of the BacMix (Fig. 4c, right panel: A II medium +sugars). The decreased but measurable expression of genes for NeuAc and/or GlcNAc catabolism suggests that *C. difficile* still had access, albeit reduced, to these sugars. This, together with access to alternative nutrient sources present in the basal medium (e.g. amino acids), may explain the increase in *C. difficile* titers over-time (from t18 to t21; Fig. 4b, right panel). Our results therefore substantiate the capability of the BacMix consortium to impact *C. difficile* growth by depleting NeuAc and/or GlcNAc.

**Targeted microbiota restoration by a consortium of *O*-glycan utilizers reduces *C. difficile* colonization levels in vivo.** The positive results obtained in vitro encouraged us to test whether the BacMix consortium would also be able to impair *C. difficile* colonization in vivo in conventional antibiotic-treated mice[42]. To this end, antibiotic-treated mice were split into two groups, where one group (BacMix group) received $5 \times 10^6$ BacMix cells by oral gavage, while the other group (control group) received the vehicle (PBS) alone (Fig. 5a). The next day, both groups were challenged with $1 \times 10^6$ *C. difficile* cells by oral gavage (Fig. 5a). Microbiota composition of both groups was monitored throughout the entire experiment by 16S rRNA gene amplicon sequencing of fecal pellets (Fig. 5b and Supplementary Fig. 5). As expected, antibiotic-treatment induced a dramatic shift in the overall microbiota composition (Fig. 5b, Bray–Curtis distance matrix, day 0 and day 6), which was marked by expansion of Enter-obacteriaceae and Burkholderiaceae (Supplementary Fig. 5a). This was accompanied by a significant decrease in alpha diversity (Supplementary Fig. 5b and c; $p = 1.602e-06$ for both Observed Richness and Shannon Index metrics, Mann–Whitney test). No significant clustering of microbiota by group could be observed at this stage. However, we did observe a significant clustering of mouse fecal microbiota by group following gavage (days 9 to 11; $p < 0.05$, PERMANOVA analyses), which can be attributed to differences in relative abundances of BacMix members such as Akkermansiaceae (*A. muciniphila*), and Peptostreptococcaceae (*C. difficile*) between the two groups (Fig. 5b and Supplementary Fig. 5a).

A closer look into the amplicon sequencing data allowed us to identify OTUs 3, 5, 11, 34 and 81 as representatives of the BacMix members *A. muciniphila*, *R. lactatiformans*, *A. timonensis*, *M. intestinale*, and *Bacteroides* sp. isolate FP24, respectively. These members were detected in the initial community (day 0) in both groups of mice at low to moderate relative abundances (from 0.1 to 9.0%) and are depleted following antibiotic treatment (day 6; Fig. 5c). All BacMix members, with the sole exception of *M. intestinale*, were restored and detected in the guts of BacMix-recipient animal at days 3 and 4 post-gavage (days 10 and 11 of

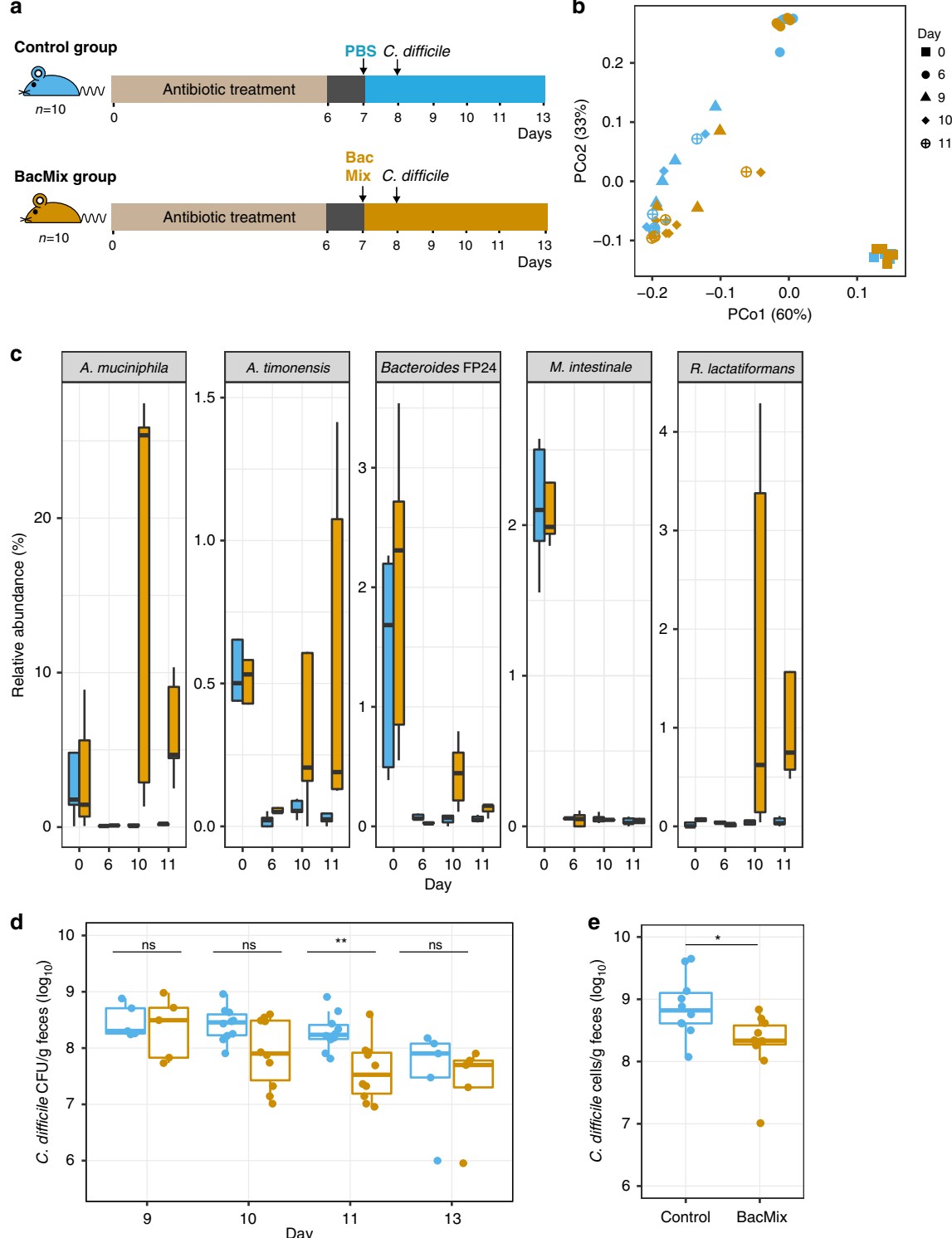

**Fig. 5 Adoptive transfer of a BacMix consortium after antibiotic exposure of conventional mice increases resistance to *C. difficile* infection.**
**a** Schematic representation of the in vivo adoptive transfer experiment. **b** Variation between samples based on 16S rRNA gene amplicon sequencing ((Principal Coordinates Analysis based on Bray–Curtis dissimilarities) colored by group and shaped by sampling day. The percentage in brackets corresponds to the variation explained by each principal coordinate (PCo) to the fraction of the total variance of the data. **c** Relative abundance of the five BacMix members in fecal samples collected from control (blue) and BacMix-recipient (brown) mice (*n* = 5/group) as inferred from 16S rRNA gene amplicon sequencing. **d**, **e** Density of *C. difficile* at the indicated time points in feces of antibiotic-treated control (blue) and BacMix-recipient (brown) mice determined by selective-medium plating (**d**) or by qPCR (**e**) (days 10 and 11: *n* = 10/group, days 9 and 13: *n* = 5/group). *p = 0.0155; **p = 0.0058; ns= non-significant; two-sided Mann–Whitney test. In (**c**–**e**), boxes represent median, first and third quartile. Whiskers extend to the highest and lowest values that are within one and a half times the interquartile range. Source data are provided as a Source Data file.

the experiment), but not in the control group (Fig. 5c). Overall, these results show that most of the BacMix members successfully colonized the gut of inoculated mice. Nevertheless, the introduction of the 5-member consortium had no impact on alpha diversity metrics, which remain the same across the two groups throughout the entire experiment (Supplementary Fig. 5b, c).

To evaluate if the BacMix members were able to impair *C. difficile* colonization, *C. difficile* levels in fecal pellets were quantified throughout the experiment by CFU enumeration of freshly collected and plated pellets and by qPCR using primers targeting the *C. difficile* 16S rRNA gene. In line with previous studies, *C. difficile* efficiently colonized the gut of antibiotic-treated animals, reaching levels of $>10^8$ CFUs per gram of feces at day 9 (1 day after infection) (Fig. 5d). No *C. difficile* could be detected in the guts of mock-infected animals handled in parallel (Supplementary Fig. 6). *C. difficile* titers in infected mice stabilized (as assessed by CFU counts) or slightly increased (as assessed by qPCR) until day 11 (day 3 post-infection) (Fig. 5d). Remarkably, we observed a gradual decrease in the titers of *C. difficile* in mice that received the BacMix compared with the control group from day 9 to day 10. This difference is only statistically significant at day 11, time point for which we detect a notable decrease in *C. difficile* titers for the BacMix group ($4.1 \pm 4.6 \times 10^7$ cells per gram of feces) compared to the control group ($2.5 \pm 2.1 \times 10^8$ cells per gram of feces) (Fig. 5d; $p = 0.0058$, Mann–Whitney test). These results were further corroborated using qPCR (Fig. 5e; $p = 0.0155$, Mann–Whitney test). At day 13 (day 5 post-infection) the titers of *C. difficile* in animals from both groups start to decrease, suggesting that pathogen clearance from the guts of these animals had already begun[32]. Furthermore, we detected a significant decrease in *C. difficile nanT* expression levels in mice from the BacMix group at day 2 post-infection (normalized to the expression of the DNA polymerase IIIC gene), indicating a reduction in the levels of available NeuAc in the guts of these animals (Fig. 4d). We therefore conclude that the BacMix consortium presented here colonizes the mouse gut of antibiotic-treated mice and is capable of reducing *C. difficile* burdens in vivo. This pathogen reduction can be explained at least in part by an impaired access to mucosal sugars, particularly to NeuAc. Importantly, we did not detect any significant impairment of *C. difficile* colonization, nor a decrease in *nanT* expression, in mice that received $5 \times 10^6$ cells of a control bacteriotherapy mix (BacMixC; composed of equal numbers of *Anaerotruncus colihominis* isolate FP23, *Lactobacillus hominis* strain DSM 23910 and of *Escherichia sp.* isolate FP11; Supplementary Table 2 and Supplementary Fig. 3) that were not detected among the RACS-sorted cells from the incubations (Supplementary Fig. 7; Supplementary Table 2).

Toxin-mediated intestinal inflammation is frequently observed in the course of a CDI, and the severity of the inflammation depends on the degree of virulence of the infecting *C. difficile* strain, among other factors[20,43]. Histopathological examination of colon sections from *C. difficile* colonized mice revealed only mild pathology and no significant differences in terms of severity between control and BacMix-recipient animals (Supplementary Fig. 8a, b). Congruent with these results, we observed low but quantifiable amounts of TcdB in *C. difficile* colonized animals (between 37.4 and 238.3 ng of TcdB per g of colon content), but no significant differences between groups (Supplementary Fig. 8c). Furthermore, we recorded only minor weight loss in our experiments (Supplementary Fig. 5d). These results are in agreement with the low levels of cytotoxic activity reported for *C. difficile* strain 630 in mice[43,44].

Enhanced mucus degradation due to an overgrowth of mucus utilizers has been reported to increase, rather than to decrease, susceptibility to infection with the gut pathogen *Citrobacter rodentium*[4]. However, we did not observe thinning of the mucus layer, nor impairment of mucin secretion in mice that received the BacMix compared to the control group (Supplementary Fig. 9). We hypothesize that the administered BacMix consortium expanded in the guts of these animals by using the pool of free NeuAc and potentially other mucosal sugars, carbohydrates or amino acids that became available after antibiotic treatment[17,45,46], rather than by over-degrading the mucus layer.

## Discussion

Experimental studies of mucin degradation by natural gut communities have addressed mucin utilization as a whole, but have not looked into the uptake of individual mucosal sugars by different taxa[34,35]. Here, we characterize for the first time the capacity of the intestinal microbiota to forage on individual mucosal sugars in the context of whole communities by exploiting a recently developed activity-based Raman-sorter[35]. The large majority of organisms identified here by single-cell deuterium labeling showed genomic evidence for the capacity to use the respective monosaccharide sugar and also became D-labeled in their proteins, together strongly supporting their roles in mucosal monosaccharide utilization. Several of the mucosal-sugar-utilizing taxa identified in this study, i.e., Muribaculaceae, *Akkermansia*, *Bacteroides* and Lachnospiraceae, have previously been identified as potential mucus degraders, but their monosaccharide preferences in the context of a complex community have not been investigated[14,34]. Our results therefore illuminate the in situ nutrient niche of key mucosal sugar utilizers. Others, such as Rikenellaceae (*Alistipes* species) (MAG RACS_006 and 026), are here reported for the first time as being directly involved in mucosal sugar uptake (Fig. 2a; Supplementary Fig. 3). Interestingly, *Alistipes* spp. have previously been proposed to live in close association with the mucus, because their levels are directly altered during epithelial inflammation[47,48]. Our results indicate *Alistipes* spp. have the capacity to utilize various mucosal derived sugars, which may therefore help to explain such observations.

A number (11 of 51) of the identified mucosal sugar foragers, i.e., Muribaculaceae, Bacteroidales, *Alistipes* and Lachnospiraceae, are generalists that actively use all or most O-glycan mucosal sugars (Fig. 2a). These organisms may be permanent mucus residents and therefore take advantage of all available mucin moieties as nutrient sources. However, the majority of the organisms here identified were only sorted from one or two of the mucosal sugars, and therefore may be more limited in their capacities to use different sugars, or partly were low in abundance and therefore might have been missed in some of the other incubations by our cell-sorting approach (Fig. 2a). Despite the high throughput of the RACS platform (up to 500 cells per hour[35]), long sorting times are not possible for technical reasons (i.e., sedimentation of cells over-time in the input microfluidics and drifting of z-plane in the optics). To avoid these two technical issues, we performed multiple 1–2 h sorting experiments for each amendment/microcosm (Supplementary Table 1). We were able to analyze on average 1000 cells per microcosm, a throughput that is two orders of magnitude higher as achieved by manual sorting[34,35], which enabled us to identify major taxa of interest, though possibly not sufficient for the recovery of extremely rare organisms.

Few sorted organisms (4 out of 51), such as Alphaproteobacteria, *Desulfovibrio* sp. and *Prevotella* sp. lack the genomic potential to catabolize the sugars amended to the microcosms they were sorted from and also tended to not be significantly enriched in the D-labeled peptide pool (Fig. 2b and Fig. 3d).

These organisms most likely became labeled and sorted due to cross-feeding of metabolites released by primary utilizers, despite the minimal incubation times employed. However, the majority of the sorted organisms had the genome-encoded potential to utilize the sugar supplemented to the microcosm from which it was sorted, with the sole exception of GalNAc (Fig. 2b). Several MAGs resulting from sorted cells from GalNAc amendments did not contain genes for canonical GalNAc catabolic enzymes[15] (Fig. 2b). However, several *Bacteroides* strains and *Akkermansia muciniphila* are able to utilize GalNAc[49,50], and also lack known genes for the canonical GalNAc catabolic pathway[15]. We therefore propose that the GalNAc-supplemented sorted organisms use an undescribed catabolic pathway for GalNAc. This finding underlies the importance of applying function-targeted approaches to identify organisms performing key functions in the gut that would be otherwise missed by random genome-centric approaches.

The function-targeted sorting of uncultured cells in this study enabled us to identify key players in NeuAc and GlcNAc catabolism, and to then rationally select a mixture of organisms (BacMix) able to impair the growth of the pathogen *C. difficile*. We observed that there was a consistent decrease of half a log in *C. difficile* levels by day 3 of infection in antibiotic-treated mice that received the BacMix compared to mice that received PBS only (Fig. 5d,e). This decrease in colonization levels detected at day 3 post-infection is comparable to colonization reductions exhibited by a *C. difficile nanT* mutant strain, which is impaired in NeuAc uptake[17]. These findings, together with the decreased expression of *C. difficile* genes required for NeuAc catabolism (Fig. 4d), suggest that mucosal sugar depletion is one of the mechanisms involved in colonization resistance. Nevertheless, we cannot rule out that mechanisms other than depletion of these sugars may also contribute and be relevant to the observed outcome. Our results show that the BacMix *per se* is not sufficient to completely halt *C. difficile* colonization, but may constitute a base for developing a more extensive combination of organisms to provide colonization resistance by additional, distinct mechanisms. For instance, the BacMix may act in synergy with organisms such as *Lactobacillus reuteri* strain 17938[28] or *Clostridium scindens*[29,30] that inhibit *C. difficile* growth either by transforming glycerol into the antimicrobial compound reuterin in the first case, or by converting primary into secondary bile acids in the second case. Such combinations may be the key to completely restore resistance to *C. difficile* colonization. It would also be relevant to further assess the impact of the BacMix alone or in combination with additional organisms in other aspects of CDI beyond colonization. The *C. difficile* 630 strain and the mouse model of CDI employed here enabled us to uncover the impact of BacMix administration on *C. difficile* colonization, but to evaluate the impact of bacteriotherapy mixtures on disease outcome, experiments with more virulent *C. difficile* strains (e.g. strain VPI 10463 or R20291) together with additional animal models of *C. difficile* pathogenesis (e.g. the hamster model of CDI[51]) are needed.

In summary, this study extends the toolbox to identify microbes within the gut microbiota with beneficial functions and provides an important basis for the development of therapeutics. Function-targeted sorting of uncultured cells enabled us to identify the key organisms involved in a specific process (mucosal sugar utilization) within a complex gut community. This information was then used to formulate a mixture of organisms that could interfere with *C. difficile* colonization, at least in part by depleting available mucosal sugars from the guts of susceptible animals. We believe that this approach provides a solid framework to design targeted bacteriotherapies for the treatment of enteric pathogens.

## Methods

**Ethics statement**. All animal experiments were performed at the Max F. Perutz Laboratories of the University of Vienna, Austria. All experiments were discussed and approved by the University of Veterinary Medicine, Vienna, Austria, and conducted in accordance with protocols approved by the Federal Ministry for Education, Science and Research of the Republic of Austria under the license number BMWF-66.006/0001-WF/V/3b/2016. Animals were randomized for interventions but researchers processing the samples and analyzing the data were aware which intervention group corresponded to which cohort of animals.

**Mouse colon incubations**. Three adult (6-8 weeks old) C57BL/6N mice bred at the Max F. Perutz Laboratories, University of Vienna, under SPF conditions were sacrificed per experiment, and their colon was harvested anaerobically (85% N₂, 10% CO₂, 5% H₂) in an anaerobic tent (Coy Laboratory Products, USA). Contents from each colon were suspended in 7.8 mL of 50% D₂O-containing PBS and homogenized by vortexing. Similar conditions have been successfully applied in the past to monitor activity of individual cells in gut communities without causing major changes in the activity of individual community members[34]. The homogenate was left to settle for 10 min, and the supernatant was then distributed into glass vials and supplemented with different concentrations of mucosal sugar monosaccharides, glucose, mucin or nothing (no-amendment control) (all amendment chemicals were from Sigma-Aldrich, except D(+)-galactose which was purchased from Carl Roth GmbH) (Fig. 1a,c). After incubation for 6 h at 37 °C, glycerol was added (to achieve a final concentration of 20% (v/v) of glycerol in the microcosms) and the vials were crimp-sealed with rubber stoppers and stored at −80 °C until further processing. Prior to glycerol addition, subsamples of the biomass were collected, pelleted and supernatants stored at −80 °C for HILIC LC-MS/MS measurements. Pellets were washed with PBS to remove D₂O and were fixed in 3% formaldehyde for 2 h at 4 °C and stored in 50% PBS/50% ethanol solution at −20 °C until further use. A total of three biological replicates were established using starting material pooled from three animals each (experiments MonoA, MonoB and MonoC). For the MonoA and MonoB experiments, microcosms were established for all the different concentrations of monosaccharides tested (Fig. 1c), while for MonoC only the highest concentrations of monosaccharides tested in incubations MonoA and MonoB were supplemented. Note that analysis of mucin-amended sorted fractions has been published elsewhere[35]. Since mucin contains all the monosaccharides included in this study, it constitutes an important control, and therefore we processed the sequencing data from mucin sorts in parallel with our samples and included it in our analyses (Fig. 2).

**Mass spectrometric analysis of mucosal monossaccharides**. Hydrophilic interaction chromatography (HILIC) LC-MS/MS was used for the measurement of mucosal monosaccharides in microcosm supernatants. Frozen samples were thawed at room temperature and centrifuged for 10 min at 18.000 × g and 4 °C. Supernatants were then diluted 1:50 with acetonitrile:water (1:1; v/v) and a volume of 3 µl was injected onto the chromatographic column. The UHPLC system (UltiMate 3000, Thermo Scientific) was coupled to a triple quadrupole mass spectrometer (TSQ Vantage, Thermo Scientific) by an electrospray ionization interface. Hydrophilic interaction chromatographic separation was realized on a Luna aminopropyl column (3 µm, 150 × 2 mm; Phenomenex, Torrance, CA) at a flow rate of 0.25 ml/min. Eluent A consisted of 95% water and 5% acetonitrile with 20 mM ammonium acetate and 40 mM ammonium hydroxide as additives and eluent B of 95% acetonitrile and 5% water. A multi-step gradient was optimized as follows: 100% B until minute 2, then linearly decreased to 80% B until minute 20 and further to 0% B until minute 25. The column was kept at 0% B for 4 min before it was equilibrated for 5 min at the initial conditions. The column temperature was maintained at 40 °C. The mass spectrometer was operated in multiple reaction monitoring (MRM) mode. Electrospray ionization (ESI) was optimized as follows: spray voltage 2800 V (positive mode) and 3000 V (negative mode); vaporizer temperature 250 °C; sheath gas pressure 30 Arb; ion sweep gas pressure 2 Arb; auxiliary gas pressure 10 Arb; capillary gas temperature 260 °C. Mass spectrometric parameters were optimized by direct injection and are reported together with the retention times of individual sugars in Supplementary Table 6. Spiking experiments and regular quality control checks were conducted to evaluate and ensure the systems' proper performance.

**Confocal Raman microspectroscopy and spectral processing of fixed samples**. Formaldehyde-fixed samples were spotted on aluminum-coated slides (Al136; EMF Corporation) and washed by dipping into ice-cold Milli-Q (MQ) water (Millipore) to remove traces of buffer components. Individual cells were observed under a 100×/0.75 NA microscope objective. Single microbial cell spectra were acquired using a LabRAM HR800 confocal Raman microscope (Horiba Jobin-Yvon) equipped with a 532-nm neodymium-yttrium aluminum garnet (Nd:YAG) laser and either 300 grooves/mm diffraction grating. Spectra were acquired in the range of 400–3200 cm⁻¹ for 30 s with 2.18 mW laser power. Raman spectra were background-corrected using the sensitive nonlinear iterative peak algorithm, and afterwards normalized to the sum of its absolute spectral intensity[34]. For quantification of the degree of D substitution in CH bonds (%CD), the bands assigned to

C–D (2040–2,300 cm$^{-1}$) and C–H (2,800–3,100 cm$^{-1}$) were calculated using integration of the specified region[34].

**Raman-activated cell sorting.** For RACS of D-labeled cells, 100 μl of glycerol-preserved microcosms containing non-fixed cells were pelleted, washed once with MQ water containing 0.3 M glycerol and finally resuspended in 0.5 ml of 0.3 M glycerol in MQ water. Cell sorting was performed in a fully automated manner using a Raman microspectroscope (LabRAM HR800, Horiba Scientific, France) combined with optical tweezers and a polydimethylsiloxane (PDMS) microfluidic sorter. The optical tweezers (1,064 nm Nd:YAG laser at 500 mW) and Raman (532 nm Nd:YAG laser at 45 mW or 80 mW; see below) laser were focused at the same position of the interface between the sample and sheath streams using a single objective (63x, 1.2 NA water-immersion, Zeiss). The in-house software based on the graphical user interface (GUI; written in MATLAB) detected the single-cell capture and its deuterium labeling status by calculating the cell index ($P_C = I_{1,620-1,670}/I_{fluid,1,620-1,670}$; where $I$ is the integrated intensity between the indicated wavenumbers) and the labeling index ($P_L = I_{2040-2300}/I_{1850-1900}$), respectively. We did not detect a significant change in the C–D peak region (2040–2300 cm$^{-1}$) due the presence of 0.3 M of glycerol in the sorting fluid (added to minimize the osmotic stress when the sample was re-suspended for the RACS) (Supplementary Fig. 2a). Other spectral regions (e.g., <1500 cm$^{-1}$ and >2700 cm$^{-1}$) were slightly affected, but the sorting algorithm employed and the parameters described above take these small changes into account: the cell index $P_C$ ($I_{1620-1670}/I_{fluid,1620-1670}$) used to detect single-cell capture was calculated by comparing the Raman intensity of cells measured in real-time to that of the working fluid measured in the calibration (conducted before the actual sorting). The threshold value for $P_L$ ($I_{2040-2300}/I_{1850-1900}$) was chosen based on the measurement of the control sample (i.e., sample incubated in non-D$_2$O-containing medium). We used two software versions, each of which uses 45 mW (version 1) and 80 mW (version 2) Raman laser powers, respectively. The second version operates with higher power based on the addition of a laser shutter that blocks the Raman laser while the cells are being translocated, reducing the laser-induced damage on the cell. This version allows shorter acquisition times to be employed, and therefore higher throughput of the platform. The laser power for each version was chosen based on visual inspection of captured cells as described[35]. For the NeuAc and GlcNAc-amendment sorts (version 1, since version 2 was not yet available), $P_C$ value was calculated from cell spectra acquired for 2 s at the "capture location", while the $P_L$ value was calculated from spectra obtained with a 5 s exposure time at the "evaluation location". Fucose, GalNAc, and galactose-supplemented sorts were performed with version 2 of the platform, which in the meantime became available, significantly reducing sorting times. For these sorts both $P_C$ and $P_L$ values were simultaneously measured at the "capture location" with a 0.3 s exposure time. Only the D-labeled cells were translocated to the 'evaluation location' and immediately released. In order to determine the threshold $P_L$ above which a cell from the microcosms should be considered D-labeled (and therefore selected and sorted), cells from glucose-supplemented microcosms incubated in the absence or presence of D (0% versus 50% D$_2$O in the microcosms) were run on the platform prior to sorting. The threshold $P_L$ number can vary across microcosms due to different microbial compositions and/or physiological status of cells present in the starting material, as well as due to different laser powers employed. Therefore we determined the $P_L$ threshold separately for both MonoA and MonoB incubations using both 45 and 80 mW laser power. Nevertheless, we reached a $P_L$ threshold of 6.19 for all sets of conditions and incubations tested (Supplementary Fig. 2b). We speculate this was due to the identical conditions used in both incubations and the fact that both communities have a similar microbial composition (Fig. 1e). To test the sorting accuracy of the platform on our samples, the negative control (H$_2$O, glucose-supplemented microcosm) was re-run in the platform and sorted using the adopted threshold ($P_L = 6.19$) (Supplementary Table 1). As expected, no cells were considered labeled by the platform under these conditions. Sorted fractions were nevertheless collected and sequenced as controls.

**Preparation of 16S rRNA gene amplicon libraries and 16S rRNA gene sequence analyses.** DNA extracted from the mouse colon microcosms or from mouse fecal pellets using a phenol-chloroform bead-beating protocol[52] was used as a template for PCR. PCR amplification was performed with a two-step barcoding approach[53]. In the first-step PCR, the 16S rRNA gene of most bacteria was targeted using oligonucleotide primers (Supplementary Table 7) containing head adaptors (5′-GCTATGCGCGAGCTGC-3′) in order to be barcoded in a second step PCR[53]. Barcode primers consisted of the 16 bp head sequence and a sample-specific 8 bp barcode from a previously published list at the 5′ end. The barcoded amplicons were purified with the ZR-96 DNA Clean-up Kit (Zymo Research, USA) and quantified using the Quant-iT PicoGreen dsDNA Assay (Invitrogen, USA). An equimolar library was constructed by pooling samples, and the resulting library was sent for sequencing on an Illumina MiSeq platform at Microsynth AG (Balgach, Switzerland). Paired-end reads were quality-filtered and processed using QIIME 1[53,54]. Reads were then clustered into operational taxonomic units (OTUs) of 97% sequence identity and screened for chimeras using UPARSE implemented in USEARCH v8.1.1861[55]. OTUs were classified using the RDPclassifier v2.12[56] as implemented in Mothur v1.39.5[57] using the Silva database v132[58]. Sequencing

libraries were rarefied and analyzed using the vegan package (v2.4-3) of the software R (https://www.r-project.org/, R 3.4.0).

**Sequencing of mouse gut isolates.** *Bacteroides* sp. Isolate FP24 was isolated from YCFA agar plates (DSMZ medium 1611- YCFA MEDIUM (modified)) by plating ten-fold dilution series of a microcosms supplemented with 2 mg/ml of NeuAc (experiment MonoB). *Escherichia* sp. isolate FP11 and *Anaerotruncus* sp. isolate FP23 were isolated from *C. difficile* minimal medium[17] agar plates supplemented with 0.25% NeuAc or 0.25% GlcNAc by plating ten-fold dilution series of a microcosms supplemented with 2 mg/ml of NeuAc (experiment MonoB) or of a microcosms supplemented with 5 mg/ml of GlcNAc (experiment MonoA), respectively. Colonies were re-streak on the same medium plates until complete purity. Pure colonies were grown overnight in 10 mL of BHI medium (Brain heart infusion at 37 g per liter of medium) supplemented with: yeast extract, 5 g; Na$_2$CO$_3$, 42 mg; cysteine, 50 mg; vitamin K1, 1 mg; hemin, 10 μg. DNA was extracted from pelleted biomass using the QIAGEN DNAeasy Tissue and Blood kit (Qiagen, Austin, TX, USA) according to the manufacturer′s instructions. Sequencing libraries were prepared using the NEBNext® Ultra™ II FS DNA kit (Illumina) and sequenced in an Illumina MiSeq platform with 300-bp paired-end sequencing chemistry (Joint Microbiome Facility, University of Vienna and Medical University of Vienna, Austria). Reads were quality trimmed with the bbduk option of BBMap (v 34.00) at phrad score 15. Quality-trimmed reads were assembled with SPAdes (v 3.11.1)[59]. For isolate FP11, assembled reads were subsequently, iteratively ($n = 6$) reassembled with SPAdes using contigs of >1 kb from the previous assembly as "trusted contigs" for input and iterating kmers from 11 to 121 in steps of 10. CheckM (v1.0.6) assessment[60] of these genomes is summarized in Supplementary Data 1.

**Mini-metagenome sequencing and genomic analyses.** Labeled RACS cells were collected into PCR tubes, lysed and subjected to whole-genome amplification using the Repli-g Single Cell Kit (QIAGEN), according to the manufacturer's instructions. Shotgun libraries were generated using the amplified DNA from WGA reactions (sorted fractions) or DNA isolated via the phenol-chloroform method (initial microcosms) as a template and Nextera XT (Illumina) reagents. Libraries were sequenced with a HiSeq 3000 (Illumina) in 2 × 150 bp mode at the Biomedical Sequencing Facility, Medical University of Vienna, Austria. The sequence reads were quality trimmed and filtered using BBMap v34.00 (https://sourceforge.net/projects/bbmap/). The remaining reads were assembled de novo using SPAdes 3.11.1[59] in single-cell mode (k-mer sizes: 21, 35, 55). Binning of the assembled reads into metagenome-assembled genomes (MAGs) was performed with MetaBAT 2 (v2.12.1)[61] using the following parameters: minContig 2000, minCV 1.0, minCVSum 1.0, maxP 95%, minS 60, and maxEdges 200. The quality and contamination of all MAGs were checked with CheckM 1.0.6[60] (Supplementary Data 1). MAGs >200 kb obtained from all samples were compared and de-replicated using dRep 1.4.3[62]. Automatic genome annotation of contigs >2 kb within each de-replicated MAG was performed with RAST 2.0[63]. Taxonomic classification of each MAG was obtained using GTDB-Tk[64] (v0.1.3, gtdb.ecogenomic.org/).

The relative abundance of each MAG on the initial microcosms was calculated based on metagenomic coverage. Filtered reads from each sequenced microcosm were mapped competitively against all retrieved MAGs using BBMap (https://sourceforge.net/projects/bbmap/). Read coverage was normalized by genome size and relative abundances of each genome in each sample were calculated based on the formula: $cov^A = (bp^A/g^A)/(bp^T/g^T)$, where $cov^A$ is the relative abundance of MAG $A$ on a particular sample, $bp^A$ is the number of base pairs from reads matching MAG $A$, $g^A$ is the genome length of MAG $A$, $bp^T$ is the total number of base pairs from reads matching all MAGs recovered from that particular sample and $g^T$ is the sum of all MAGs genome lengths.

For determination of the presence of encoded enzymes for catabolism of mucin monosaccharides among MAGs, predicted protein sequences from recovered MAGs were subject to local BLASTP analyses[65], against a custom database. The database was composed of all enzymes involved initial hydrolysis and catabolism of mucosal sugar monosaccharides (Supplementary Data 2), which were previously curated from a total of 395 human gut bacteria[15]. A strict e-value threshold of 10$^{-50}$ was used for all BLASTP analyses. During initial setup of the analysis pipeline, functional assignments of proteins that gave positive BLASTP hits were manually verified by examining annotations from RAST 2.0 and by performing BLASTP analyses against the NCBI-nr database (NCBIblast 2.2.26).

To verify the enrichment of a selected dataset of mucin-degrading enzymes[35] in the assemblies derived from sorted fractions, BLASTX analyses of scaffolds from each fraction as well as from the initial microcosms metagenomes (unsorted) were performed against the selected mucin-degrading enzyme sequences[15] (Supplementary Table 3). An e-value threshold of 10$^{-50}$ was also used for all BLASTX analyses.

**Phylogenomic analyses.** A concatenated marker alignment of 34 single-copy genes was generated for all MAGs using CheckM 1.0.6[60] and the resulting alignment was used to calculate a tree with the approximate maximum-likelihood algorithm of FastTree 2.1.10[66]. Phylogenomic trees were visualized and formatted using iTOL v4 (https://itol.embl.de/). In order to identify the closest relative for

each MAG, the query MAG and close reference genomes (based on the generated phylogenomic tree) were compared using dRep 1.4.3[62]. Compared genomes with a whole-genome-based average nucleotide identity (ANIm) >99%[39] were considered to be the same population genome.

**High-resolution mass spectrometric analyses.** Glycerol-preserved biomass (150 μL) from microcosm incubations was pelleted and suspended in 50 μL of lysis buffer (1% sodium dodecyl sulfate (SDS), 10 mM TRIS base, pH 7.5). Protein lysates were subjected to SDS polyacrylamide gel electrophoresis followed by in-gel tryptic digestion. Proteins were stained with colloidal Coomassie Brilliant Blue G-250 (Roth, Kassel, Germany) and detained with Aqua dest. Whole protein bands were cut into gel pieces and in-gel-digestion with trypsin 30 μL (0.005 μg/μL) was performed overnight. Extracted peptides where dried and resolved in 0.1% formic acid and purified by ZipTip® treatment (EMD Millipore, Billerica, MA, USA).

In total, 5 μg of peptides were injected into nanoHPLC (UltiMate 3000 RSLCnano, Dionex, Thermo Fisher Scientific), followed by separation on a C18-reverse phase trapping column (C18 PepMap100, 300 μm × 5 mm, particle size 5 μm, nano viper, Thermo Fischer Scientific), followed by separation on a C18-reverse phase analytical column (Acclaim PepMap® 100, 75 μm × 25 cm, particle size 3 μm, nanoViper, Thermo Fischer Scientific). Mass spectrometric analysis of eluted peptides where performed on a Q Exactive HF mass spectrometer (Thermo Fisher Scientific, Waltham, MA, USA) coupled with a TriVersa NanoMate (Advion, Ltd., Harlow, UK) source in LC chip coupling mode. LC Gradient, ionization mode and mass spectrometry mode were performed as described before[67]. Briefly, peptide lysate were injected into a Nano-HPLC and trapped in a C18-reverse phase column (Acclaim PepMap® 100, 75 μm × 2 cm, particle size 3 μM, nanoViper, Thermo Fisher) for 5 min. Peptide separation was followed by a two-step gradient in 90 min from 4 to 30% of B (B: 80% acetonitrile, 0.1% formic acid in MS-grade water) and then 30 min from 30 to 55% of B. The temperature of the separation column was set to 35 °C and the flow rate was 0.3 μL/min. The eluted peptides were ionized and measured. The MS was set to a full MS/dd-MS[2] mode scan with positive polarity. The full MS scan was adjusted to 120,000 resolution, the automatic gain control (AGC) target of $3 \times 10^6$ ions, maximum injection time for MS of 80 s and a scan range of 350 to 1550 $m/z$. The dd-MS[2] scan was set to a resolution of 15,000 with the AGC target of $2 \times 10^5$ ions, a maximum injection time for 120 ms, TopN 20, isolation window of 1.6 $m/z$, scan range of 200 to 2000 $m/z$ and a dynamic exclusion of 30 s.

Raw data files were converted into mzML files and searched with MS-GF+ against a database obtained from microcosm metagenomes composed of 276,284 predicted protein-encoding sequences. The following parameters were used for peptide identification: enzyme specificity was set to trypsin with one missed cleavage allowed using 10 ppm peptide ion tolerance and 0.05 Da MS/MS tolerance. Oxidation (methionine) and carbamidomethylation (cysteine) were selected as modifications. False discovery rates (FDR) were determined with the node Percolator[68]. Proteins were considered as identified when at least one unique peptide passed a FDR of 5%.

The *MetaProSIP* toolshed[69] embedded in the Galaxy framework[70] (v2.3.2, http://galaxyproject.org/) was used to identify the incorporation of stable isotopes into peptides. *MetaProSIP* calculates the relative isotope abundance (RIA) on detected isotopic mass traces ($m/z$ tolerance of ±10 ppm, intensity threshold of 1000, and an isotopic trace correlation threshold of 0.7).

**In vitro growth experiments.** *A. muciniphila* strain Muc (DSM 22959), *Ruthenibacterium lactatiformans* strain 585-1 (DSM 100348) and *Alistipes timonensis* strain JC136 (DSM 25383) were obtained from DSMZ. *Muribaculum intestinale* strain YL27 (DSM 28989) was kindly provided by Prof. Bärbel Stecher (Max-von-Pettenkofer Institute, LMU Munich, Germany). *Bacteroides* sp. FP24 was isolated from YCFA agar plates (DSMZ medium 1611-YCFA MEDIUM (modified)). All strains were grown in reduced A II medium[71] consisting of (per liter of medium): BHI, 18.5 g; yeast extract, 5 g; trypticase soy broth, 15 g; $K_2HPO_4$, 2.5 g; hemin, 10 μg; glucose, 0.5 g; $Na_2CO_3$, 42 mg; cysteine, 50 mg; menadione, 5 μg; fetal calf serum (complement-inactivated), 3% (vol/vol). For *A. muciniphila* cultivation, the growth medium was supplemented with 0.025% (w/v) of mucin. *C. difficile* was grown in BHI medium (37 g per liter of medium) supplemented with: yeast extract, 5 g; $Na_2CO_3$, 42 mg; cysteine, 50 mg; vitamin K1, 1 mg; hemin, 10 μg. All strains were grown at 37 °C under anaerobic conditions until stationary phase, and then serially diluted and plated into media agar plates in order to determine the number of viable cells present in 1 ml of stationary phase-culture. For mixed-growth experiments, the culture volume equivalent to $1 \times 10^6$ CFU of each strain was pelleted, cells were washed with PBS, mixed in equal proportions and finally resuspended in 100 μl of PBS. This bacterial mixture containing a total of $5 \times 10^6$ BacMix cells was then used to inoculate 2.5 ml of A II medium (diluted two fold in 2× PBS) supplemented or not with 0.25% (10 mM) carbon source (0.125% or 4 mM of NeuAc and 0.125% or 6 mM of GlcNAc). After 12 h, the same tube was inoculated with $1 \times 10^6$ *C. difficile* CFU and bacterial growth was followed by measuring the OD at 600 nm every hour until stationary phase. At three distinct points of the *C. difficile* growth curve—lag (t12, right after *C. difficile* addition), mid-exponential (t18) and early stationary phase (t21)—a sample aliquot was collected and ten-fold dilutions were plated in a *C. difficile* selective medium[72]. This selective medium (CCFA) includes antibiotics such as cycloserine and cefoxitn at

concentrations that are inhibitory to most gut organisms, except for *C. difficile*, allowing to determine total *C. difficile* counts. A second aliquot was immediately pelleted and the pellet was stored at −80 °C for RNA extraction.

**Quantitative PCR of *C. difficile* 16S rRNA gene copy number density.** DNA was extracted from 100 mg of mouse fecal pellet using the QIAGEN DNAeasy Tissue and Blood kit (Qiagen, Austin, TX, USA) according to the manufacturer's instructions, with an additional step of mechanical cell disruption by bead beating (30 s at 6.5 m/s) right after addition of kit lysis buffer AL. Extracted DNA (2 μl) was subjected to quantitative PCR using 0.2 μM of primers specifically targeting the *C. difficile* 16S rRNA gene[73] (Supplementary Table 7) and 1× SYBR green Master Mix (Bio-Rad) in a total reaction volume of 20 μl. Standard curves were generated from DNAs extracted from fecal pellets of SPF (uninfected) mice spiked in with different known numbers of *C. difficile* cells ($10^2$, $10^3$, $10^4$, $10^5$, $10^6$, $10^7$, and $10^8$) as described in Kubota et al., 2014. Amplification and detection were performed using a CFX96™ Real-Time PCR Detection System (Bio-Rad) using the following cycling conditions: 95 °C for 5 min, followed by 40 cycles of 95 °C for 15 s, 56 °C for 20 s, and 72 °C for 30 s. To determine the specificity of PCR reactions, melt curve analysis was carried out after amplification by slow cooling from 95 to 60 °C, with fluorescence collection at 0.3 °C intervals and a hold of 10 s at each decrement. Only assays with amplification efficiencies above 80% were considered for analysis.

**RNA extraction and quantitative real-time PCR.** Total nucleic acids (TNA) were extracted from mouse fecal pellets or from in vitro cultures using a phenol-chloroform bead-beating protocol[52]. RNA was purified from DNAse-treated TNA fractions using the GeneJET Cleanup and Concentration micro kit (Thermo Fisher Scientific). cDNA was synthesized from 0.5 μg of total RNA with 1 μl of random hexamer oligonucleotide primers. Samples were heated for 5 min at 70 °C. After a slow cooling, 2 μl of deoxynucleoside triphosphates (dNTP; 2.5 mM each), 40 units of recombinant ribonuclease inhibitor (RNaseOUT) and 4 μl of reverse transcription (RT) buffer were added and cDNAs were synthesized for 2 h at 50 °C using 200 units SuperScript™ III Reverse Transcriptase (all reagents used in cDNA synthesis were from Thermo Fisher Scientific). Real-time quantitative PCR was performed in a 20-μl reaction volume containing 2 μl of cDNA, 1x SYBR green Master Mix (Bio-Rad) and 0.2 μM of gene-specific *C. difficile* primers targeting the following genes: DNA polymerase III PolC-type *dnaF*[74], *nanA*, *nanT*[17] and *nagB* (this work; Supplementary Table 7). Amplification and detection were performed as described above. In each sample, the quantity of cDNAs of a gene was normalized to the quantity of cDNAs of the *C. difficile* DNA polymerase lII gene[74] (*dnaF*). The relative change in gene expression was recorded as the ratio of normalized target concentrations (threshold cycle [ΔΔCT] method[75]). Fold changes were normalized to in vitro growths in *C. difficile* minimal medium containing 0.5% glucose[17]. To determine the specificity of PCR reactions, melt curve analysis was carried out after amplification by slow cooling from 95 to 60 °C, with fluorescence collection at 0.3 °C intervals and a hold of 10 s at each decrement. Only assays with amplification efficiencies above 80% were considered for analysis.

**Murine in vivo adoptive transfer experiments.** Female C57BL/6N 6-8 weeks old mice ($n = 33$ total) were purchase from Janvier Labs. Animals were kept in isolated, ventilated cages under specific pathogen-free conditions at the animal facility of the Max F. Perutz Laboratories, University of Vienna, Austria, with controlled temperature of 21 ± 1 °C and humidity of 50 ± 10%, in a 12-h light/dark cycle. Mice received a standard diet (V1124-300; Ssniff, Soest, Germany) and autoclaved water ad libitum. Mice were administered antibiotics (0.25 mg/ml clindamycin (Sigma-Aldrich) for six days in drinking water) and subsequently assigned randomly to one of two groups. One day following antibiotic cessation, mice from each group were split into 3 cages (to minimize the cage effect) and each mouse received either 5,000,000 CFU of a 5-bacteria suspension (BacMix, containing equal numbers of *A. muciniphila* strain Muc (DSM 22959), *Ruthenibacterium lactatiformans* strain 585-1 (DSM 100348), *Alistipes timonensis* strain JC136 (DSM 25383), *Muribaculum intestinale* strain YL27 (DSM 28989), and *Bacteroides* sp. isolate FP24) or vehicle (PBS) by gavage (Fig. 5a). At the time of BacMix and BacMixC administration, the mouse diet was switched from a standard diet (V1124-300; Ssniff, Soest, Germany) to a isocaloric polysaccharide-deficient chow[76] with sucrose but no cellulose or starch (Ssniff, Soest, Germany). For the BacMixC adoptive transfer, each mouse ($n = 10$ per group) received 5,000,000 CFU of a 3-bacteria suspension (BacMixC, containing equal numbers of *Anaerotruncus* sp. isolate FP23; *Lactobacillus hominis* strain DSM 23910 and of *Escherichia* sp. isolate FP11) or vehicle (PBS) by gavage (Supplementary Fig. 7). One day after BacMix or BacMixC administration, mice were challenged with 1,000,000 CFU of *C. difficile* strain 630 delta*Erm*[77]. *A. muciniphila*, *R. lactatiformans* and *M. intestinale* were grown in reduced A II medium (supplemented with 0.025% mucin for *A. muciniphila*). *Anaerotruncus* sp. isolate FP23, *A. timonensis* and *Bacteroides* sp. isolate FP24 were grown in PYG (DSMZ medium 104). *Lactobacillus hominis* and *Escherichia* sp. isolate FP11 were grown in YCFA medium (DSMZ medium 1611). *C. difficile* was grown in BHI medium (37 g per liter of medium). All bacteria were grown under anaerobic conditions (5% $H_2$, 10% $CO_2$, rest $N_2$) at 37 °C and resuspended in anaerobic PBS prior to administration to animals. *C. difficile* titers were quantified in fecal samples obtained from mice 24, 48, 72, and 120 h after infection by overnight cultivation in

*C. difficile* selective agar plates[72]. Animals were monitored throughout the entire experiment and weight loss was recorded.

**Measurement of mucus thickness and goblet cell volume**. Segments of mouse colon (approximately 10 mm long) were fixed in 2% PFA in phosphate-buffered saline (PBS) for 12 h at 4 °C. Samples were washed with 1× PBS and then stored in 70% ethanol at 4 °C until embedding. For embedding, samples were immersed in 3 changes of xylene for 1 h each, then immersed in 3 changes of molten paraffin wax (Paraplast, Electron Microscopy Sciences) at 56–58 °C for one hour each. Blocks were allowed to harden at room temperature. Sections were cut to 4 μm thickness using a Leica microtome (Leica Microsystems) and were floated on a water bath at 40–45 °C, then transferred to slides, dried, and stored at room temperature. In preparation for staining, slides were de-paraffinized by heating at 60 °C for 30 min followed by immersion in 2 changes of xylene for 5 min each, then in 2 changes of 100% ethanol for 1 minute each, then rehydrated through 80%, and 70% ethanol for 1 minute each. Slides were then dipped in water, drained, air-dried and a drop of Alcian Blue stain (Sigma Aldrich) applied on top. Samples were incubated with Alcian Blue for 20 min at room temperature and then washed in water to remove the excess of stain. Samples were mounted in Vectashield Hardset™ Antifade Mounting Medium (Vector Laboratories) and visualized using a Leica Confocal scanning laser microscope (Leica TCS SP8X, Germany). For measurements of the width of the mucus layer and determination of goblet cell volume per crypt, ImageJ software (https://imagej.nih.gov/ij/, 1.48 v) was used.

**Histology and histopathological scoring**. Mouse colons were flushed with cold PBS to remove all contents, and the entire colon was rolled into Swiss rolls. The tissues were fixed in 2% PFA for 12 h at 4 °C and then washed in 1x PBS and transferred to 70% ethanol until embedding. Swiss rolls were embedded in the same manner as described above for segments of mouse colon. Paraffin-embedded, PFA fixed tissues were sectioned at 4.5 μm. Tissue sections were de-paraffinized and hematoxylin and eosin stained (Mayer's Hematoxylin, Thermo Scientific; Eosin 1%, Morphisto, Germany). Histopathological analyses were performed using a semi-quantitative scoring system[78] that evaluated the severity of crypt damage and cellular infiltration, epithelial erosion and tissue thickening using a severity score from 0 to 3 (0 = intact, 1 = mild, 2 = moderate, 3 = severe), and those scores were multiplied by a score for percent involvement (0 = 0%, 1 = 1–25%, 2 = 26–50%, 3 = 50–100%). A trained and blinded scientist performed the scoring. Representative images were acquired using an Olympus CKX53 microscope and Olympus SC50 camera.

**Quantification of *C. difficile* toxin TcdB**. Levels of TcdB in mouse colon contents were quantified relative to a standard curve of purified TcdB using an ELISA assay kit ("Separate detection of *C. difficile* toxins A and B", TGC Biomics) according to the manufacturer's instructions. For each mouse, approximately 10 mg of colon content were used in the assay. The limit of detection for the assay in our conditions was determined to be 5.14 ng of TcdB per gram of colon content (Supplementary Fig. 8c). One of the mice from the BacMix group had toxin levels below the detection limit and was therefore excluded from analysis.

**Reporting summary**. Further information on research design is available in the Nature Research Reporting Summary linked to this article.

## Data availability
The data that support the findings of this study are available from the corresponding author upon request. 16S rRNA gene sequence data have been deposited in the NCBI Sequence Read Archive (https://www.ncbi.nlm.nih.gov/sra) under SRP226368. Metagenomic data have been deposited in the NCBI Sequence Read Archive under SRP227836 and SRP144778 (for mucin amendment metagenomic data[35]). RACS MAGs have been deposited as whole-genome shotgun projects at DDBJ/ENA/GenBank (https://www.ncbi.nlm.nih.gov/genbank/) under the accessions WSLP00000000-WSNN00000000 (Supplementary Table 8). The mass spectrometry proteomic data has been deposited to the ProteomeXchange Consortium (http://proteomecentral.proteomexchange.org) via the PRIDE[79] partner repository with the dataset identifier PXD015215. Source data are provided with this paper.

## Code availability
All of the custom codes used in this study can be accessed upon reasonable request from the corresponding author. MATLAB GUI (graphical user interface) and code used in the RACS sorting (version 1; RACS_ver.1) is available at https://github.com/harubang2/MATLAB-platform-for-Raman-activated-cell-sorting-RACS. MATLAB GUI (graphical user interface) and code used in the RACS sorting (version 2) is available at https://github.com/fatimapereira454/MATLAB-platform-for-Raman-activated-cell-sorting-RACS-version2.

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

## Acknowledgements

This work was supported by the Austrian Science Fund (FWF; P27831-B28, P26127-B20, and ZK-57), the European Union's Horizon 2020 Framework Programme for Research and Innovation (grant No.658718 to FCP), and the European Research Council (Starting Grant: FunKeyGut 741623). Martin von Bergen is grateful for funding by the grant from the Deutsche Forschungsgesellschaft in the framework of the CRC 1382 Gut-Liver-Axis: Functional Circuits and Therapeutic Targets. We thank Michaela Lang and Anita Krnjic, as well as staff members of the animal facility of the Max F. Perutz Laboratories, Vienna,

for technical assistance. We are grateful to Christos Zioutis for help with the proteomics data analysis. We are grateful to Petra Pjevac and the Joint Microbiome Facility, Vienna, Austria, for support with sequencing and sequence data analysis. We thank Alexander Loy for helpful discussions and valuable comments.

## Author contributions

F.C.P. and D.B. conceived and designed the experiments. F.C.P. and B.S. performed the experiments. F.C.P. and D.B. performed the data analysis and wrote the manuscript. B.W. and I.K. performed the mass spectrometric analysis of mucosal monossaccharides. N.J. and M.B. performed the high-resolution mass spectrometric analyses. K.W. and C.W.H. performed bioinformatic analyses. C.V. performed the histopathological analysis. R.S. and K.S.L. supported the RACS platform. T.D. supported the animal experiments. M.W. provided critical analysis and input. All authors have given approval to the final version of the paper.

## Competing interests

The authors declare no competing interests.
