## [Peer Review File · Nature Communications]

Response to Reviewers

Manuscript NCOMMS-19-30845003

Rational design of a microbial consortium of mucosal sugar utilizers reduces *Clostridiodes difficile* colonization

Pereira, FC, Wasmund, K, Cobankovic, I, Jehmlich, N, Herbold, CW, Lee, KS, Sziranyi, B, Vesely, C, Decker, T, Stocker, R, von Bergen, M, Wagner, M and Berry, D.

We thank all four Reviewers for their valuable comments, which we have used to improve the manuscript.

In the following pages, we have reproduced the Reviewers' comments verbatim and provide point-by-point responses and a description of the modifications that we have made to the manuscript. All answers to Reviewers' comments are highlighted in blue. All modifications of the manuscript's original text are highlighted in yellow on the revised version.

REVIEWERS' COMMENTS

Reviewer #1 (Remarks to the Author):

The paper by Pereira et al uses a functional approach to identify members of the mouse microbiota that can metabolize different carbohydrates associated with the host epithelium and mucus. The first half of the paper describes a novel approach to not only identify sugar metabolizing strains but to also retrieve them and yield a metagenomic profile. Overall the first half of the paper is well-presented and establishes a new way to find bugs that metabolize not only these sugars but presumably any carbon/nitrogen source that can be metabolized by bacteria. The second half of the paper describes a competition between organisms that can metabolize the sugars used in this study (however not from any of the strains identified in the first half of the paper). Here the data are mostly circumstantial and the authors do not provide any direct evidence that the competition for sugars is what is leading to the modest decrease in *C. difficile* numbers in their model. Overall this approach to identify individual members of the microbiome that metabolize these mucus associated sugars is very interesting, however the strong conclusions that the BacMix developed here need more experimental evidence to support them or need to be softened to note that other mechanisms are possible based on the data presented.

We appreciate the reviewer's constructive comments and suggestions, which we have used to improve the manuscript.

Specific comments.

1. The competition experiments presented in figure 4 aren't really competitions. The BacMix was added several hours prior to the *C. difficile* strain. To establish this is a direct competition one would need to add them together and see if the BacMix is more efficient at eating these sugars than *C. difficile*. As tested – the BacMix essentially have eaten nearly all of the sugar before *C. difficile* was added, so the result obtained was the only outcome possible.

The reviewer raises a valid point about the specific language used, and we agree that the experiment does not reflect a direct "competition". We have therefore removed the term "competition" when referring to this experiment throughout the manuscript. We also clarify that our results rather suggest "depletion" of NeuAc and GlcNAc (rather than direct competition for NeuAc and GlcNAc) as one of the mechanisms through which the BacMix is able to decrease *C. difficile* levels.

Indeed in preliminary experiments, our attempts to establish a direct competition between *C. difficile* and the BacMix failed due to the fact that we could not identify a culture medium that can sustain *in vitro* growth of all 6 strains at similar growth rates. That is, *C. difficile* grows faster than any of the BacMix strains in all liquid media we have tested (BHI, YCFA, A II; **Supplementary Fig. 4**), both in the absence or presence of the added sugars. We speculate this is because distinct organisms require different proportions of micro- and macro-nutrients for optimal growth, and possibly also because *C. difficile* can efficiently use nutrients such as amino acids for energy conservation.

We believe these *in vitro* experiments nevertheless helped us to rule out a possible negative outcome, i.e., the BacMix strains may not have been able to efficiently grow when mixed, or grow but not act in synergy to efficiently deplete the sugars under the conditions tested, e.g., they could have inhibited each other. This was not the case and therefore we believe that this is an important experiment as it provided us with the first

evidence that the BacMix strains can act in synergy to decrease the access of *C. difficile* to the sugars NeuAc and GlcNAc *in vitro*.

Ultimately, this is in line with our *in vivo* experiments and approach in which the BacMix was added before *C. difficile*, in order to prevent pathogen expansion in a prophylactic manner.

The specific changes are as follows:

Lines 111-116: “We show that this bacterial consortium is able to decrease the availability of these mucosal sugars and can reduce *C. difficile* growth both *in vitro* and *in vivo*. Our approach therefore identified a consortium of gut bacteria that contributes to colonization resistance against *C. difficile*, and further indicates that depletion of mucosal sugars is one of the mechanisms underpinning this resistance”

Lines 285-286 (section head): “Depletion of NeuAc and GlcNAc by selected commensals reduces *C. difficile* growth *in vitro*”

Lines 291-293: “These observations prompted us to investigate whether commensal NeuAc and GlcNAc utilizers identified here could efficiently deplete these mucosal sugars, and by doing so, reduce *C. difficile* growth and/or colonization.”

Lines 496-501: “These findings, together with the decreased expression of *C. difficile* genes required for NeuAc catabolism (Figure 4d), suggest that mucosal sugar depletion is one of the mechanisms involved in colonization resistance. Nevertheless, we cannot rule out that mechanisms other than depletion of these sugars may also contribute and be relevant to the observed outcome.”

2. In the animal experiment, CD630 was used as the *C. difficile* strain and is one that causes only mild disease. The strains chosen for the BacMix are known to metabolize these sugars, however it is possible that other mechanisms of reducing *C. difficile* numbers by the modest five fold reduction are also possible. At the very least the authors should show that a strain mix incapable of metabolizing these sugars provides no colonization resistance to *C. difficile*. It is likely genetics are not possible in these strains but if there were strains available that have a deletion of gene(s) involved in these pathways that would be a much better test of the hypothesis. As it stands now the only evidence that *C. difficile* is seeing less of these sugars is the reduction in NanT expression, which based on the error bar seems to be highly variable. Please present these data with the individual points rather than bar charts so readers can get a better sense of the variation.

We agree that using a modified version of the BacMix incapable of sugar metabolism would be ideal to show that utilization of GlcNAc and NeuAc is the sole and/or main mechanism behind the observed colonization resistance provided by the BacMix. However, as the reviewer mentioned, there are no tools available at the moment that enable such an experiment. Our results suggest that depletion of these mucosal sugars by the BacMix is one of the reasons behind the differences we observe in *C. difficile* colonization (**Figure 4d**). We have now modified the manuscript text and title to clearly communicate that our results suggest that utilization/depletion of NeuAc and GlcNAc is a mechanism behind the partial restoration of colonization resistance observed, but it may not be the only mechanism at play, as follows:

Manuscript title: “Rational design of a microbial consortium of mucosal sugar utilizers reduces *Clostridiodes difficile* colonization”

Lines 111-116: “We show that this bacterial consortium is able to decrease the availability of these mucosal sugars and can reduce *C. difficile* growth both *in vitro* and *in vivo*. Our approach therefore identified a consortium of gut bacteria that contributes to

colonization resistance against *C. difficile*, and further indicates that depletion of mucosal sugars is one of the mechanisms underpinning this resistance”

Lines 496-501: “These findings, together with the decreased expression of *C. difficile* genes required for NeuAc catabolism (Figure 4d), suggest that mucosal sugar depletion is one of the mechanisms involved in colonization resistance. Nevertheless, we cannot rule out that mechanisms other than depletion of these sugars may also contribute and be relevant to the observed outcome.”

We have also modified **Figures 4c** and **4d** and we now show all the individual data points behind each bar.

3. Presumably the authors also measured *C. difficile* numbers before and after day 3. These data would be useful to see in interpreting the change at day 3.

We did measure *C. difficile* titers by plating and counting the colony forming units in the faeces of colonized animals before and after day 3 of infection, more precisely at days 1, 2, 3 and 5 post-infection (respectively days 9, 10, 11 and 13 of the experiment). We have now included this data in **Figure 5d**. We did observe a gradual decrease in the titers of *C. difficile* in mice that received the BacMix compared with the control group from day 9 to day 10, but this difference is only statistically significant at day 11 (day 3 post-infection). At day 13 (day 5 post-infection) we could still observe a similar trend, but less pronounced, as overall the titers of *C. difficile* in mice from both groups began to decrease. This is a sign of pathogen clearance from the guts of these animals, which is known to happen in this model and has been reported in other studies (Hryckowian et al., 2018).

We also discuss this data on the manuscript, as follows:

Lines 380-395: “In line with previous studies, *C. difficile* efficiently colonized the gut of antibiotic-treated animals, reaching levels of $>10^8$ CFUs per gram of feces at day 9 (1 day after infection) (Figure 5d). No *C. difficile* could be detected in the guts of mock-infected animals handled in parallel (Supp. Figure 6). *C. difficile* titers in infected mice stabilized (as assessed by CFU counts) or slightly increased (as assessed by qPCR) until day 11 (day 3 post-infection) (Figure 5d). Remarkably, we observed a gradual decrease in the titers of *C. difficile* in mice that received the BacMix compared with the control group from day 9 to day 10. This difference is only statistically significant at day 11, time point for which we detect a notable decrease in *C. difficile* titers for the BacMix group ($4.1 \pm 4.6 \times 10^7$ cells per gram of feces) compared to the control group ($2.5 \pm 2.1 \times 10^8$ cells per gram of feces) (Figure 5d; $p=0.005$, Mann-Whitney test). These results were further corroborated using qPCR (Figure 5e; $p=0.013$, Mann-Whitney test). At day 13 (day 5 post-infection) the titers of *C. difficile* in animals from both groups start to decrease, suggesting that pathogen clearance from the guts of these animals had already begun³².”

4. Because *C. difficile* 630 is so mild in its disease causing capabilities, the authors may consider using a more virulent strains (R20291 or VPI) to see if the BacMix has an effect on weight loss, survival, or toxin production in addition to numbers.

We are aware that strain 630 used causes only moderate disease in the mouse CDI model, and we agree that it would be relevant in the future to also look at the effect of the BacMix on infection outcome using a more virulent *C. difficile* strain. Nevertheless, our primary goal was to better understand the role of the identified mucosal sugar utilizers in resistance against *C. difficile* colonization, from an ecological perspective. We chose 630 as it is a well characterized strain and it has been widely used in studies aimed at

understanding the mechanisms employed by *C. difficile* to colonize the gut, as well as the role of the gut microbiota in colonization resistance (Janoir et al., 2013; Ferreyra et al., 2014, Ng et al., 2013; Hryckowian et al., 2018; Battaglioli et al., 2018). Furthermore, we have reasons to believe that the outcome of our experiments in terms of *C. difficile* colonization levels may not drastically differ when using a more virulent strain, as it has been shown that virulent (BI1, VPI 10463) and “mild” (630, F200) *C. difficile* strains attain very similar colonization levels, despite the different levels of cytotoxicity and disease severity reported (Theriot et al., 2011).

We have now included a sentence in the manuscript discussion that reflects this answer, as follows:

Lines 509-516: “It would also be relevant to further assess the impact of the BacMix alone or in combination with additional organisms in other aspects of CDI beyond colonization. The *C. difficile* 630 strain and the mouse model of CDI employed here enabled us to uncover the impact of BacMix administration on *C. difficile* colonization, but to evaluate the impact of bacteriotherapy mixtures on disease outcome, experiments with more virulent *C. difficile* strains (e.g strain VPI 10463 or R20291) together with additional animal models of *C. difficile* pathogenesis (e.g. the hamster model of CDI⁵¹) are needed.”

5. The title is problematic for two reasons. One, the consortium doesn't provide much of an effect on *C. difficile* numbers and I suspect that the differences must be only for day 3 since no other days are presented. Thus, it is not a big effect. Second, the authors haven't shown it is the sugar degrading capability of any member of the BacMix that is providing the modest effect.

We agree with the reviewer and we have now modified the manuscript title. It now reads: “Rational design of a microbial consortium of mucosal sugar utilizers reduces *C. difficile* colonization”

References cited:

- Janoir, C., Denève, C., Bouttier, S., Barbut, F., Hoys, S., Caleechum, L. et al. (2013) Adaptive Strategies and Pathogenesis of Clostridium difficile from In Vivo Transcriptomics. Infection and Immunity 81: 3757-3769.
- Hryckowian, A.J., Van Treuren, W., Smits, S.A., Davis, N.M., Gardner, J.O., Bouley, D.M., and Sonnenburg, J.L. (2018) Microbiota-accessible carbohydrates suppress Clostridium difficile infection in a murine model. Nat Microbiol 3: 662-669.
- Ferreyra, J.A., Wu, K.J., Hryckowian, A.J., Bouley, D.M., Weimer, B.C., and Sonnenburg, J.L. (2014) Gut microbiota-produced succinate promotes *C. difficile* infection after antibiotic treatment or motility disturbance. Cell host & microbe 16: 770-777.
- Ng, K.M., Ferreyra, J.A., Higginbottom, S.K., Lynch, J.B., Kashyap, P.C., Gopinath, S. et al. (2013) Microbiota-liberated host sugars facilitate post-antibiotic expansion of enteric pathogens. Nature 502: 96.
- Theriot, C.M., Koumpouras, C.C., Carlson, P.E., Bergin, II, Aronoff, D.M., and Young, V.B. (2011) Cefoperazone-treated mice as an experimental platform to assess differential virulence of Clostridium difficile strains. Gut Microbes 2: 326-334.
- Battaglioli, E.J., Hale, V.L., Chen, J., Jeraldo, P., Ruiz-Mojica, C., Schmidt, B.A. et al. (2018) Clostridioides difficile uses amino acids associated with gut microbial dysbiosis in a subset of patients with diarrhea. Sci Transl Med 10.

- Hutton, M.L., Mackin, K.E., Chakravorty, A., and Lyras, D. (2014) Small animal models for the study of *Clostridium difficile* disease pathogenesis. *FEMS microbiology letters* **352**, 140-149.

Reviewer #2 (Remarks to the Author):

In the manuscript titled "Rational design of a microbial consortium able to outcompete *Clostridiodes difficile* for key sugars" by Fátima C. Pereira and colleagues identified mouse gut bacteria that utilize specific mucus-derived monosaccharides using single-cell stable isotope-probing, Raman-activated cell sorting and metagenomics. The authors identify members of the family Muribaculaceae, in addition to Lachnospiraceae, Rikenellaceae, and Bacteroidaceae as major utilizers of mucin monosaccharides which are preferred nutrients for *C. difficile*. The authors have done a phenomenal job of methodically identifying bacteria that can out-compete *C. difficile* for the open nutrient niches using innovative techniques. The authors have to be commended on the effort to sequence as well as assemble the genomes, confirming sugar utilization using gene expression studies and the extensive *in vitro* and *in vivo* validation. The authors assembled a five-member consortium of sialic acid and N47 acetylglucosamine utilizers using the above approach which has the potential to replace FMT as a treatment modality for *C. difficile* infection. Overall this is an exciting story; I have two small concerns as below

We appreciate the reviewer's supportive words and comments, which we have used to improve our manuscript.

1. The authors should consider including a control 5 member consortium of bacteria which are similar to the assembled consortium of sialic acid and N47 acetylglucosamine utilizers in other respects but unable to utilize the sugars in their *in vitro* and *in vivo* studies. This will help determine the specificity of the 5 member community in terms of excluding *C. difficile*. It will be hard to get genetic controls but a control community can help address concerns about specificity.

We agree that using a genetically modified version of the BacMix incapable of sugar metabolism would be the only direct way to elucidate if utilization of these sugars is or not the sole and main mechanism behind the observed colonization resistance provided. In the absence of available tools that would allow such an experiment to be performed, we have modified the manuscript text and title to clarify that our results suggest that utilization of NeuAc and GlcNAc is a mechanism behind the partial restoration of colonization resistance provided by the BacMix, but it may not be the only or even the main mechanism at play. It now reads as follows:

Manuscript title: "Rational design of a microbial consortium of mucosal sugar utilizers reduces *Clostridiodes difficile* colonization"

Lines 111-116: "We show that this bacterial consortium is able to decrease the availability of these mucosal sugars and can reduce *C. difficile* growth both *in vitro* and *in vivo*. Our approach therefore identified a consortium of gut bacteria that contributes to colonization resistance against *C. difficile*, and further indicates that depletion of mucosal sugars is one of the mechanisms underpinning this resistance"

Lines 496-501: "These findings, together with the decreased expression of *C. difficile* genes required for NeuAc catabolism (Figure 4d), suggest that mucosal sugar depletion is one of the mechanisms involved in colonization resistance. Nevertheless, we cannot rule out that mechanisms other than depletion of these sugars may also contribute and be relevant to the observed outcome."

Regarding the specificity of the BacMix in reducing *C. difficile* levels in the gut, results from the adoptive transfer of the control community BacMixC (**Supplementary Fig. 7** and **Supplementary Table 3**; community of organisms that were present in the microcosms, but that were not recovered from the sorted fractions) show that contrary to the BacMix, the BacMixC does not significantly impact on *C. difficile* colonization levels, nor does it lead to a decrease in the expression of *nanT* in *C. difficile* (**Supplementary Fig. 7e**). This is mentioned in the manuscript, **lines 402-408**: “Importantly, we did not detect any significant impairment of *C. difficile* colonization, nor a decrease in *nanT* expression, in mice that received 5×10^6 cells of a control bacteriotherapy mix (BacMixC; composed of equal numbers of *Anaerotruncus colihominis* isolate FP23, *Lactobacillus hominis* strain DSM 23910 and of *Escherichia sp.* isolate FP11; Supp. Table 2 and Supp. Figure 3) that were not detected among the RACS-sorted cells from the incubations (Supp. Figure 7; Supp. Table 3).”

2. The authors primarily report colonization data but they should also provide data on toxin production and histopathology as *C. difficile* colonization is not the same as infection. *C. difficile* can colonize without causing pathology; the question is if nutrient competition prevents *C. difficile* infection rather than just colonization.

We now provide histopathology analysis as well as quantification of toxin (TcdB) levels in the gut of *C. difficile*-colonized animals from the Control and BacMix-recipient group (**Supplementary Fig. 8**). Histopathological examination of colon sections stained with hematoxylin and eosin revealed only mild pathology and no significant differences in terms of disease severity between the two groups (**Supplementary Figure 8a and b**). In agreement with these results, we observed low but quantifiable amounts of TcdB in *C. difficile* colonized animals, and only a small decrease between TcdB levels of BacMix-recipient mice when compared to the Control group (**Supplementary Figure 8c**). In agreement with the histopathology results, we also detected minimal weight loss in infected mice, and this loss was similar in Control and BacMix groups (**Supplementary Figure 5d**). We believe these results may be due to: 1) the modest impact of the BacMix adoptive transfer on *C. difficile* total titers, and/or 2) that the *C. difficile* strain used (630) has been reported to cause only mild disease in mice, with only minimal to no histopathological changes and low cytotoxicity observed in gut tissues of infected animals (Theriot *et al.*, 2011).

We agree that it would be relevant in the future to also look at the effect of the BacMix on infection outcome when using a virulent *C. difficile* strain able to cause more severe disease. However, we have reasons to believe that the outcome in terms of *C. difficile* colonization levels may not drastically differ if using a more virulent strain, as it has been shown that virulent (BI1, VPI 10463) and “mild” (630, F200) *C. difficile* strains attain very similar colonization levels, despite the different levels of cytotoxicity and disease severity reported (Theriot *et al.*, 2011).

We report the histopathological results and discuss the relevance of assessing disease outcome, as follows:

Lines 409-419: “Toxin-mediated intestinal inflammation is frequently observed in the course of a *C. difficile* infection, and the severity of the inflammation depends on the degree of virulence of the infecting *C. difficile* strain, among other factors^{20,43}. Histopathological examination of colon sections from *C. difficile* colonized mice revealed only mild pathology and no significant differences in terms of severity between Control and BacMix-recipient animals (Supp. Figure 8a and b). Congruent with these results, we observed low but quantifiable amounts of TcdB in *C. difficile* colonized animals (between

37.4 and 238.3 ng of TcdB per g of colon content), but no significant differences between groups (Supp. Figure 8c). Furthermore, we recorded only minor weight loss in our experiments (Supp. Figure 5d). These results are in agreement with the low levels of cytotoxic activity reported for *C. difficile* strain 630 in mice^{43,44}.”

Lines 509-516: “It would also be relevant to further assess the impact of the BacMix alone or in combination with additional organisms in other aspects of CDI beyond colonization. The *C. difficile* 630 strain and the mouse model of CDI employed here enabled us to uncover the impact of BacMix administration on *C. difficile* colonization, but to evaluate the impact of bacteriotherapy mixtures on disease outcome, experiments with more virulent *C. difficile* strains (e.g strain VPI 10463 or R20291) together with additional animal models of *C. difficile* pathogenesis (e.g. the hamster model of CDI⁵¹) are needed.”

References cited:

- Theriot, C.M., Koumpouras, C.C., Carlson, P.E., Bergin, II, Aronoff, D.M., and Young, V.B. (2011) Cefoperazone-treated mice as an experimental platform to assess differential virulence of *Clostridium difficile* strains. *Gut Microbes* **2**: 326-334.
- Reeves, A.E., Koenigskecht, M.J., Bergin, I.L., and Young, V.B. (2012) Suppression of *Clostridium difficile* in the gastrointestinal tracts of germfree mice inoculated with a murine isolate from the family Lachnospiraceae. *Infect Immun* **80**: 3786-3794.
- Hutton, M.L., Mackin, K.E., Chakravorty, A., and Lyras, D. (2014) Small animal models for the study of *Clostridium difficile* disease pathogenesis. *FEMS microbiology letters* **352**, 140-149.
- Rupnik, M., Wilcox, M.H., and Gerding, D.N. (2009) *Clostridium difficile* infection: new developments in epidemiology and pathogenesis. *Nat Rev Microbiol* **7**: 526-536.

Reviewer #3 (Remarks to the Author):

Review comments on

Ref.: NCOMMS-19-30845003

Title: Rational design of a microbial consortium able to outcompete *Clostridiodes difficile* for key sugars

by Berry et al.

This is an excellent work to apply Raman activated cell sorting to study mouse gut microbiome. It is very interesting that authors managed to restrain the growth of pathogen *C. difficile* using BacMix to compete essential sugar. Raman analysis and RACS methodology are based on the authors' recent papers, which are convincing and statistical data analysis is sound. The authors validate that the incorporation into RACS-sorted taxa using SIP metaproteomics, which is a very good foundation for future Raman sorting based in deuterium incorporation.

We appreciate the reviewer's supportive words and comments, which we have used to improve our manuscript.

However, there are a few questions as follows.

1. In RACS, cross-feeding of deuterium would potentially be an issue, so the sampling time is important. Why t=6 hour incubation time was chosen (line 524)? Have the fecal communities incubating at 50% D2O was sampled at other time-points around 6 hours such as t=0 h, <6h and >6 h?

We agree with the reviewer. We had indeed performed a test experiment in which the colon community was incubated with glucose (positive control) and with fucose (as a representative of a mucosal sugar), in the presence of 50% D₂O, for 0, 4, 6 and 18 hours (**Supplementary Fig. 1**, included in the Revised manuscript version). After 4 hours of incubation we do observe that a large percentage (64%) of cells have already incorporated D to detectable levels in response to glucose, but not to fucose, which may be related with different pathways and energy gained from the catabolism of these sugars. Levels of D incorporation (%CD) and percentage of labeled cells increase by hour 6 for glucose-amended microcosms, and we also do see a substantial increase in the number of cells responding to fucose (30%). At a later time-point (t18), levels of D incorporation as well as percentage of D labeled cells remain stable or increase only slightly. On the other hand, for this late time point we observe an increase in the percentage of labeled cells in the negative control (no amendment), mostly likely due to the stimulation of cells by compounds released from cell death and lysis in nutrient-deprived conditions. These results suggest that at later time points cross-feeding does indeed occur, which could interfere with our assay. We therefore selected t6 time point for our experiments as there was sufficient D labeling to be detected by Raman in response to both the positive control and the mucosal sugars and minimal background activity.

We have included a sentence in the Results section mentioning why we chose 6 hours as the incubation time:

Lines 124-127: “Based on a preliminary experiment to determine the most suitable time point to efficiently probe for D incorporation, an incubation period of 6 hours was selected for all subsequent incubations (Supp. Figure 1).”

We have also included a detailed explanation in the legend of **Supplementary Figure 1**, as follows:

“After 4 hours of incubation we observed that a large percentage (64%) of cells had incorporated D to detectable levels in response to glucose, but not to fucose, which may be related with different pathways and energy gained from the catabolism of the two sugars. Levels of D incorporation (%CDs) as well as the percentage of labeled cells increased by hour 6 in both microcosms. At a later time point (18 hours), levels of D incorporation as well as percentage of D labeled cells remained stable or increased only slightly, but we do observe an increase in the percentage of labeled cells in the negative control (no amendment), most likely due to cross-feeding. We therefore selected 6 hours as the incubation time for our experiments, as there was enough D labeling to be detected by Raman in response to both the positive control and mucosal sugar, while cross-feeding at this time point still appeared to be minimal.”

2. Would glycerol interfere with Raman detection in RAC due to its contribution to Raman background?

We thank the reviewer for commenting on this point. We have measured the Raman spectra of four media using the RACS platform: MilliQ, 0.2M and 0.3M glycerol (balanced with MilliQ), and LB. This data is now included in the manuscript (**Supplementary Fig. 2a**). There was not a significant change in the C–D peak region (2,040–2,300 cm⁻¹) due to addition of glycerol (used to minimize the osmotic stress when the sample was re-suspended for the RACS), whereas other spectral regions (e.g., <1,500 cm⁻¹ and >2,700 cm⁻¹) were more affected (**Supplementary Fig. 2a**). Please note that our sorting algorithm can take these small changes into account: $P_C (I_{1,620-1,670} / I_{fluid,1,620-1,670})$ was calculated by comparing the Raman intensity of cells measured in real time to that of the working fluid measured in the calibration conducted before the actual sorting; and the

threshold value for P_L ($I_{2,040-2,300} / I_{1,850-1,900}$) was chosen based on the measurement of the control sample (i.e., sample incubated in non-D₂O-containing medium). In contrast to what we observe for glycerol, working fluids such as LB do generate a strong fluorescence background that veils the C–D peak intensity and should be completely avoided as the software cannot properly calculate the P_L values under these conditions (**Supplementary Fig. 2a**).

This information is now included in the manuscript in **Supplementary Fig. 2a**, as well as in the text: **Lines 618-628**: “We did not detect a significant change in the C–D peak region (2,040–2,300 cm⁻¹) due the presence of 0.3M of glycerol in the sorting fluid (added to minimize the osmotic stress when the sample was re-suspended for the RACS) (Supp. Figure 2a). Other spectral regions (e.g., <1,500 cm⁻¹ and >2,700 cm⁻¹) were slightly affected, but the sorting algorithm employed and the parameters described above take these small changes into account: the cell index P_C ($I_{1,620-1,670} / I_{\text{fluid},1,620-1,670}$) used to detect single cell capture was calculated by comparing the Raman intensity of cells measured in real-time to that of the working fluid measured in the calibration (conducted before the actual sorting). The threshold value for P_L ($I_{2,040-2,300} / I_{1,850-1,900}$) was chosen based on the measurement of the control sample (i.e., sample incubated in non-D₂O-containing medium).”

Line 143, did RACS sort fixed cells? If so, how the genome be recovered from the fixed cells?

For RACS sorting we have used only non-fixed, glycerol-preserved cells, as represented on the scheme of **Figure 1a**. We have now made this clear in the Materials and Methods section and in the main manuscript text:

Lines 181-183: “In order to identify the organisms involved in mucin O-glycan foraging, we sorted non-fixed D-labeled cells from the supplemented microcosms using RACS and shotgun-sequenced DNA from collected cells (Supp. Figure 2c).”

Lines 605-608: “For Raman-activated cell sorting of D labeled cells, 100µl of glycerol-preserved microcosms containing non-fixed cells were pelleted, washed once with MQ water containing 0.3 M glycerol and finally resuspended in 0.5 ml of 0.3 M glycerol in MQ water.”

3.Line 584, what is the power of 532-nm laser for the Raman detection in RACS? Would the sorted cells from RACS be still alive? Please comment on it.

For the RACS, we used two versions that operate with different laser powers (45 mW or 80 mW), as described in detail in the manuscript Materials and Methods, lines 742–791. For readers’ better understanding, this information has been moved to the sentence that elucidates the power of the optical tweezers laser (**line 611**). It now reads “The optical tweezers (1,064 nm Nd:YAG laser at 500 mW) and Raman (532 nm Nd:YAG laser at 45 mW or 80 mW; see below) laser were focused at the same position ...”.

Regarding the cell viability after the RACS, in our last publication (Lee *et al.*, 2019), we demonstrated that *Marinobacter adhaerens* cells grow on agar plate after RACS, implying that the short laser exposure during the RACS does not induce significant photophoretic damage to cells (**Accompanying Fig. 1**; adopted from Supplementary Fig. 7 in our last publication; Lee *at al.*, 2019). We also attained cultivation of mucin-degrading microbes after RACS, either on agar plate or in liquid medium (**Accompanying Fig. 2**), as long as the RACS operation takes less than 45 minutes, in order to minimize stress induced by exposure of cells to oxygen. Unlike the cultivation of

pure bacterial cultures (e.g., *M. adhaerens*), we did not quantify the recovery efficiency of gut microbiota sorted cells because many microbes cannot be cultured using standard media and conditions. The cultivation of sorted and/or novel gut microbes requires further investigation.

Accompanying Fig. 1 | Evaluation of the recovery of live cells. **a**, Flow chart for the quantification of the recovery of live cells by the RACS platform using *Marinobacter adhaerens* cells, CFP and deuterium-labelled as a sample. After 1 h of RACS, the collected cells were spread on an agar plate and cultured overnight. Fluorescent colonies (**b**) were counted to determine the recovery efficiency. Dividing counts of growing cells by the number of cells recorded as ‘collected’ by the RACS program yielded a recovery efficiency of viable cells of $81.8 \pm 5.9\%$. This is representative of similar results seen in three independent experiments.

Accompanying Fig. 2 | Gut bacterial cells are viable and can be cultured after RACS. Cells that were collected during RACS were cultured 48 h or 24 h on an YCFA agar plate or in YCFA medium (shown side by side with a control sample of YCFA medium containing no cells on the left), respectively.

4.Line 190-197, any estimation of the percentage of the genes were recovered from the sorted metagenomics? Please comment that the metagenomics sequences from the small population without jeopardizing the recovery of essential genes in the sorted bacteria.

We believe the reviewer’s question is related to the completeness level of the genomes recovered using RACS. We present this information on **Figure 2** and **Supp. Table 2** under “Completeness/ Contamination” and “completeness level”, respectively. We have assessed the completeness of the recovered genomes using CheckM (Parks et al.,

2015). As we mention in the manuscript, **lines 187-190**: the recovered “MAGs represented a total of 51 unique population genomes (as defined by sharing an average nucleotide identity³⁹, ANI < 99%), including 9 near-complete genomes and 24 substantially-complete genomes (Supp. Table 2).”

The remaining recovered genomes (18) have levels of completeness between 50% and 70%. We are aware that some genes may have been missed due to the fact that these are incomplete genomes, as we state in the manuscript **lines 232-234**: “Despite the fact that retrieved MAGs consisted of incompletely-reconstructed genomes, we could identify complete or near-complete catabolism pathways (with the exception of GalNAc) in many (48%) of the MAG-monosaccharide combinations.” Nevertheless we believe this does not halt our analyses/conclusions as we still see enrichment for mucosal sugar catabolism pathways on genomes from sorted fractions (**Supp. Table 4**).

5. Figure 4. How to count the number of *C. difficile* which was mixed with BacMix 326 consortium?

We have determined *C. difficile* numbers in the growth mixture by plating a sample aliquot in a *C. difficile* selective medium agar plate (Tyrrel et al., 2013). This medium includes two antibiotics, cefoxitin and cycloserine, at concentrations to which *C. difficile* is naturally resistant, but the BacMix strains are not, and therefore colonies formed on these plates belong to *C. difficile*. We have added this information to the Materials and Methods section:

Lines 824-825: “This selective medium (CCFA) includes antibiotics such as cycloserine and cefoxitin at concentrations that are inhibitory to most gut organisms, except for *C. difficile*, allowing to determine total *C. difficile* counts.”

It is interesting to see *C. difficile* was significantly lower at the time point t18 (Figure 4b in the case of A II medium + sugar), can the authors discuss why *C. difficile* significantly increase at t21? At t21, was the difference between *C. difficile* and BacMix significant?

The levels of *C. difficile* in the presence of the BacMix are indeed significantly lower than in the absence of the BacMix for t18 ($p < 0.01$, Welch two sample *t* test), but this difference is not so prominent at a later time point (**Figure 4b**, right panel: A II medium+sugars). There is still a difference at t21, but is no longer statistically significant ($P = 0.051$; Welch two sample *t* test), as explained in the manuscript, **lines 316-320**. We believe the increase in *C. difficile* levels observed from t18 to t21 can be partly explained by the consumption of mucosal sugars that were not used by the BacMix. Data shown on **Figure 4c** partially supports this hypothesis, as the expression of *C. difficile* genes involved in the catabolism of NeuAc and GlcNAc is reduced when the BacMix is present, but not completely abolished (**Figure 4c**, right panel: A II medium+sugars). This means *C. difficile* can still partially access some of the added sugars and use them to expand. Additionally, *C. difficile* may be also be growing on other nutrients sources present in the basal medium (A II). Overall, our results show a clear impact on *C. difficile* growth owing to the presence of the BacMix, and that *C. difficile* access to sugars is significantly reduced.

We better explain this results in the manuscript, as follows:

Lines 332-338: “The decreased but measurable expression of genes for NeuAc and/or GlcNAc catabolism suggests that *C. difficile* still had access, albeit reduced, to these sugars. This, together with access to alternative nutrient sources present in the basal medium (e.g. aminoacids), may explain the increase in *C. difficile* titers over time (from t18 to t21; Figure 4b, right panel). Our results therefore substantiate the capability of our BacMix consortium to impact *C. difficile* growth by depleting NeuAc and/or GlcNAc.”

It would be helpful to keep the scale of Y axis in Figure 4b and 4c consistent, e.g. 0-100.

We believe that the important measures to be directly compared are the *C. difficile* titers in CFU/mL within **Figure 4b** (left panel and right panel). These are already on the same scale. We hope the reviewer agrees that Figures 4b and 4c represent very different types of data (gene expression for three different *C. difficile* genes at t18 in **Figure 4c**, or *C. difficile* CFU/mL at 3 different time points in Figure 4b) and that normalization would not make much sense in these circumstances. Furthermore, the data on **Figure 4c** is already normalized to the expression of a *C. difficile* housekeeping gene (DNA PolIII), as described in Materials and Methods, **line 866**.

6. In the section: "Targeted microbiota restoration by a consortium of O-glycan utilizers reduces *C. difficile* colonization levels in vivo "

Line 349-353, would bacteria survive and reach the gut when they were introduced by oral gavage?

Since 16S-rRNA sequencing detects the DNA, would 16S-rRNA analysis in the fecal pellets reflect the population of actual surviving BacMix bacteria which were introduced by oral gavage?

16S-rRNA sequencing could pick up DNA from the dead BacMix present in the fecal pellets. So Line 374-375, according to 16S-rRNA sequencing alone and Figure 5, it is difficult to convince that "Overall, our results show that most of the BacMix members successfully colonized the gut of inoculated mice".

Although *C. difficile* somehow has been restrained, it can not be ruled out the dead BacMix has the effect, as a recent report suggested that the dead bacteria should have a treatment impact in the gut (<https://pubmed.ncbi.nlm.nih.gov/32277872/>).

For the same reason, it is hard to draw the conclusion in Line 399-401.

The reviewer raises a valid point. During gavage, anaerobic bacteria are exposed to high levels of oxygen and to the acidic pH of the stomach, and many do not survive these harsh conditions. The DNA from cells that reach the intestine but are no longer viable can still be recovered and amplified by 16S rRNA amplicon sequencing of mouse faeces, but only for a limited interval of time following gavage. Importantly, we expect all debris from dead cells to be cleared from the gut within 1-2 days following gavage (or even less), given the fast intestinal transit time of mice (usually 6 to 7 hours; Padmanabhan et al., 2013). Still, oral gavage is a widely applied procedure to introduce organisms in the gastrointestinal tract (Nebendhal, 2000; Theriot et al., 2011). It is known that a fraction of the large number of gavaged cells (10^6 cells) survive and reach the large intestine, where they find suitable conditions to resume growth and expand in the gut. We have reasons to believe this is the case in our study because: 1) the data from **Figure 5c** shows that gavaged organisms can be detected at stable levels at days 3 and 4 post-gavage (days 9 and 10 of the experiment), a time at which any dead cells would have already been cleared from the mouse gut; 2) they are only detected in the guts of BacMix-recipient animals, but not control animals, ruling out a spontaneous recovery of the native mouse microbiota at these late time points; 3) we detect all BacMix members at days 9 and 10, except *Muribaculum intestinale*. If we would be amplifying the 16S rRNA gene from dead cells, we should be also amplifying DNA from death cells from *M. intestinale*, which was also gavaged. This is not the case and we believe that *M. intestinale* is not detected because it was not able to establish itself in the gut, or at least not to levels we could detect by amplicon sequencing.

We modified the manuscript text to better explain our results:

Lines 366-372: “These members were detected in the initial community (day 0) in both groups of mice at low to moderate relative abundances (from 0.1 to 9.0%) and are depleted following antibiotic treatment (day 6; Figure 5c). All BacMix members, with the sole exception of *M. intestinale*, were restored and detected in the guts of BacMix-recipient animal at days 3 and 4 post-gavage (days 10 and 11 of the experiment), but not in the control group (Figure 5c). Overall, these results show that most of the BacMix members successfully colonized the gut of inoculated mice.”

References cited:

- Tyrrell, K.L., Citron, D.M., Leoncio, E.S., Merriam, C.V., and Goldstein, E.J.C. (2013) Evaluation of cycloserine-cefoxitin fructose agar (CCFA), CCFA with horse blood and taurocholate, and cycloserine-cefoxitin mannitol broth with taurocholate and lysozyme for recovery of *Clostridium difficile* isolates from fecal samples. *Journal of clinical microbiology* **51**: 3094-3096.
- Lee, K. S., Palatinszky, M., Pereira, F. C., Nguyen, J., Fernandez, V. I., Mueller, A. J., Menolascina, F., Daims, H., Berry, D., Wagner, M. & Stocker, R. (2019) An automated Raman-based platform for the sorting of live cells by functional properties. *Nature Microbiology* **4**, 1035–1048
- Theriot, C.M., Koumpouras, C.C., Carlson, P.E., Bergin, II, Aronoff, D.M., and Young, V.B. (2011) Cefoperazone-treated mice as an experimental platform to assess differential virulence of *Clostridium difficile* strains. *Gut Microbes* **2**: 326-334.
- Parks, D.H., Imelfort, M., Skennerton, C.T., Hugenholtz, P., and Tyson, G.W. (2015) CheckM: assessing the quality of microbial genomes recovered from isolates, single cells, and metagenomes. *Genome Res* **25**: 1043-1055.
- Nebendhal C. (2000) Routes of administration, p 463–483 : Krinke GJ. The laboratory rat. London (UK): Academic Press.
- Padmanabhan, P., Grosse, J., Asad, A.B.M.A., Radda, G.K., and Golay, X. (2013) Gastrointestinal transit measurements in mice with ^{99m}Tc-DTPA-labeled activated charcoal using NanoSPECT-CT. *EJNMMI Research* **3**: 60.

Reviewer #4 (Remarks to the Author):

Key results/Originality and significance:

This work used D2O coupled Raman active cell sorting and subsequently, mini-metagenomes to identify organisms capable of utilizing monosaccharide sugar from intestinal microbiota. A number of key mucosal sugar utilizers have been identified. With these discoveries, the authors demonstrated a novel approach to create probiotic mixtures for combating *Clostridioides difficile* pathogen colonization.

The work is novel and demonstrates the significant advantage of the proposed approach for isolating unknown, low abundance functional organisms from complex communities as well as identification of metabolically associated organisms – all these could have been missed by conventional random genomic approaches. Therefore, the work will be of great interest to a range of communities.

We appreciate the reviewer’s supportive words and comments, which we have used to improve our manuscript.

Minor issues in methodology & statistics:

In this work, “Between 125 and 244 cells” were sorted for each amendment. These numbers seem low since this work aims to identify key functional individual cells from the whole community. The Raman activated cell sorter is said to operate in an automated

mode. Therefore, it is difficult to understand why only hundreds of cells were sorted. The authors should explain the rationale behind this.

The RACS platform enabled us to analyze on average 1,000 cells per microcosm/amendment (Supp. Table 1), a throughput that is two orders of magnitude higher than what could be previously achieved using manual sorting (Berry et al., 2015). Despite the high throughput of the RACS platform (up to 500 cells per hour; Lee et al., 2019), sorting for long periods of time is not possible for two technical reasons. First, as the density of microorganisms is greater than that of the working fluid, cell sedimentation occurs as cells flow from a syringe into microfluidic tubing and then into a microfluidic device. As a consequence, the cell capture frequency in the optical tweezers decreases with time. Second, the microscope stage drifts in the z-direction over time. This drift is a general phenomenon and commercial microscope manufacturers now provide an add-on module to avoid it. For both cell index (P_C) and labeling index (P_L) calculations performed by the platform we chose spectral regions that are not largely affected by the measurement position in the z-direction. Still, after long sorting periods, the Raman signal from materials of the microfluidic device (glass and pdms) increases due to drift, and interference may occur. To overcome these technical issues, we performed multiple 1–2 h sorting experiments for each amendment/microcosm. We predict the RACS throughput can be increased in the future by adopting technical improvements that enable the platform to operate continuously for longer periods of time. Nevertheless, the current set up enabled us to identify major taxa involved in mucosal sugar utilization, as validated using D-metaproteomics.

We have now included a sentence in the manuscript discussing the throughput as follows:

Lines 461-469: “Despite the high throughput of the RACS platform (up to 500 cells per hour³⁵), long sorting times are not possible for technical reasons (*i.e.*, sedimentation of cells over-time in the input microfluidics and drifting of z-plane in the optics). To avoid these two technical issues, we performed multiple 1–2 hour sorting experiments for each amendment/microcosm (Supp. Table 1). We were able to analyze on average 1,000 cells per microcosm, a throughput that is two orders of magnitude higher as achieved by manual sorting^{34,35}, which enabled us to identify major taxa of interest, though possibly not sufficient for the recovery of extremely rare organisms.”

The Raman activated cell sorter played an important role in discovering the key functional organisms, but there is limited information about its operation. The readers were directed to Lee’s paper in 2019. However, there are several differences between this work and Lee’s work.

- Firstly, the rationale for classifying a D-labelled cell in this work has changed to PL/PL-threshold. It is not clear why.

In our work, we used the same rationale to determine if a cell is labeled as described by Lee et al., 2019. The P_L/P_L threshold shown in Supp. Figure 1c of the previous manuscript version was just a different way we chose to represent the data. However, we agree that this new representation may generate some confusion, and therefore we now present the same data in the format originally presented by Lee et al., 2019 (**Supplementary Fig. 2c**). Additionally, we have clarified several aspects of the RACS operation (including how was the threshold P_L determined) in the Materials and Methods section of the revised version of the manuscript, where it now reads:

Lines 635-642: “For the NeuAc and GlcNAc-amendment sorts (version 1, since version 2 was not yet available), P_C value was calculated from cell spectra acquired for 2

seconds at the 'capture location', while the P_L value was calculated from spectra obtained with a 5 second exposure time at the 'evaluation location'. Fucose, GalNAc, and galactose-supplemented sorts were performed with version 2 of the platform, which in the meantime became available, significantly reducing sorting times. For these sorts both P_C and P_L values were simultaneously measured at the 'capture location' with a 0.3 second exposure time."

Lines 643-655: "In order to determine the threshold P_L above which a cell from the microcosms should be considered D labeled (and therefore selected and sorted), cells from glucose-supplemented microcosms incubated in the absence or presence of D (0% versus 50% D_2O in the microcosms) were run on the platform prior to sorting, as described in Lee et al., 2019. The threshold P_L number can vary across microcosms due to different microbial compositions and/or physiological status of cells present in the starting material, as well as due to different laser powers employed. Therefore we have determined the P_L threshold separately for both MonoA and MonoB incubations using both 45 and 80 mW laser power. Nevertheless, we reached a P_L threshold of 6.19 for all sets of conditions and incubations tested (**Supp. Figure 2b**). We speculate this was due to the identical conditions used in both incubations and the fact that both communities have a similar microbial composition (Figure 1e)."

Also, PL-threshold is from the controls without D_2O , indicating the PL-threshold might need to be determined for each microcosm? Therefore, a clear justification is required.

The reviewer is correct; the P_L threshold number can vary across microcosms due to different microbial compositions and/or physiological status of cells present in the starting material. We have determined the P_L threshold for both MonoA and MonoB incubations using two different laser powers, and we reached the same value (P_L threshold=6.19; **Supplementary Figure 2b**). We do not find this surprising given the identical conditions used in both incubations and the fact that both communities are very similar on their microbial composition. We now represent on **Supplementary Figure 2b** all the controls analyzed and the threshold P_L value reached for all conditions. We now clearly explain it in the Materials and Methods section, **Lines 643-655** transcribed in our previous answer.

- Secondly, Lee's paper has shown a clear Raman signal of MQ water in the range between 2040 and 2300 cm^{-1} . It is highly likely that 0.3M glycerol solution used in this work may have higher Raman signals. The authors should include these background spectra in the Supplementary Figure 1 and discuss their influences on the PL calculation & identification of a D-labelled cell.

We thank the Reviewer for raising such an important point. The answer to this question is included in a previous answer; please refer to point 2 of Reviewer #3.

Reference cited:

- Lee, K. S., Palatinszky, M., Pereira, F. C., Nguyen, J., Fernandez, V. I., Mueller, A. J., Menolascina, F., Daims, H., Berry, D., Wagner, M. & Stocker, R. (2019) An automated Raman-based platform for the sorting of live cells by functional properties. *Nature Microbiology* **4**, 1035–1048

REVIEWERS' COMMENTS

We would like to thank again all four Reviewers for their valuable comments and feedback.

Reviewer #1 (Remarks to the Author):

I commend the authors for a well presented response to not only my review but all four reviews. I also appreciate the addition of animal data that does not necessarily support the main conclusion but does give readers an understanding of the impact of BacMix on *C. difficile* in vivo and in vitro. The rewording of the significance of these findings is very well done and in line with the data.

Robert Britton

Reviewer #2 (Remarks to the Author):

all questions have been addressed. In light of the lack of differences in histological changes, it is appropriate to tone down the conclusions and keep the caveat that the results may be strain specific and may not apply across the spectrum of all clinical isolates.

We agree with the Reviewer. As we mention in the discussion of the manuscript, it would be relevant in the future to assess the impact of the BacMix on colonization and infection by *C. difficile* strains other than the one tested in this work.

Lines 600-605: "The *C. difficile* 630 strain and the mouse model of CDI employed here enabled us to uncover the impact of BacMix administration on *C. difficile* colonization, but to evaluate the impact of bacteriotherapy mixtures on disease outcome, experiments with more virulent *C. difficile* strains (e.g strain VPI 10463 or R20291) together with additional animal models of *C. difficile* pathogenesis (e.g. the hamster model of CDI⁵¹) are needed."

Reviewer #3 (Remarks to the Author):

The revised manuscript has addressed all of my concerns.

Reviewer #4 (Remarks to the Author):

The revised version has improved substantially. I support it for publication.